# Structure of a Wbl protein and implications for NO sensing by *M. tuberculosis*

Bassam K. Kudhair[1,2], Andrea M. Hounslow[1], Matthew D. Rolfe[1], Jason C. Crack[3], Debbie M. Hunt[4], Roger S. Buxton [4], Laura J. Smith[1,5], Nick E. Le Brun [3], Michael P. Williamson[1] & Jeffrey Green[1]

*Mycobacterium tuberculosis* causes pulmonary tuberculosis (TB) and claims ~1.8 million human lives per annum. Host nitric oxide (NO) is important in controlling TB infection. *M. tuberculosis* WhiB1 is a NO-responsive Wbl protein (actinobacterial iron–sulfur proteins first identified in the 1970s). Until now, the structure of a Wbl protein has not been available. Here a NMR structural model of WhiB1 reveals that Wbl proteins are four-helix bundles with a core of three α-helices held together by a [4Fe-4S] cluster. The iron–sulfur cluster is required for formation of a complex with the major sigma factor ($\sigma^A$) and reaction with NO disassembles this complex. The WhiB1 structure suggests that loss of the iron–sulfur cluster (by nitrosylation) permits positively charged residues in the C-terminal helix to engage in DNA binding, triggering a major reprogramming of gene expression that includes components of the virulence-critical ESX-1 secretion system.

[1] Molecular Biology and Biotechnology, University of Sheffield, Sheffield S10 2TN, UK. [2] Department of Laboratory Investigations, Faculty of Science, University of Kufa, Najaf 54001, Iraq. [3] Centre for Molecular and Structural Biochemistry, School of Chemistry, University of East Anglia, Norwich Research Park, Norwich NR4 7TJ, UK. [4] Division of Mycobacterial Research, MRC National Institute for Medical Research, London NW7 7UH, UK. [5] School of Pharmacy, De Montfort University, Leicester LE1 9BH, UK. Nick E. Le Brun, Michael P. Williamson and Jeffrey Green jointly supervised this work. Correspondence and requests for materials should be addressed to J.G. (email: jeff.green@sheffield.ac.uk)

The actinobacteria have long been of interest because many species are medically or economically significant[1]. Thus, members of the Mycobacteriacae are the causative agents of global diseases such as tuberculosis (*Mycobacterium tuberculosis*) and leprosy (*Mycobacterium leprae*); the Actinomycetales include *Streptomyces*, which are the main source of currently used clinical antibiotics (e.g., *Streptomyces griseus* and *Streptomyces orientalis*); and members of the Corynebacterinacae are important in biotechnology as producers of amino acids (e.g., *Corynebacterium glutamicum*) and also as infectious agents (*Corynebacterium diphtheriae*)[2–5]. Actinobacteria have several distinguishing characteristics, including the presence of genes encoding white B-like (Wbl) proteins. Wbl proteins were discovered in *Streptomyces coelicolor* and were named from the appearance of mutant bacterial colonies that were impaired in the production of aerial spores[6]. Other Wbl proteins have roles in antibiotic production, antibiotic resistance, and cell division and are therefore implicated in bacterial developmental processes. All Wbl proteins possess four conserved cysteine residues within the N-terminal region and a predicted DNA-binding motif toward the C terminus[7]. The first Wbl protein to be characterized was WhiD from *S. coelicolor*, which proved to be an iron–sulfur protein with a nitric oxide- (NO-) sensitive [4Fe-4S] cluster[8–10].

*M. tuberculosis* is the causative agent of human tuberculosis (TB), which claimed 1.8 million lives in 2015[11]. A key component of TB pathogenesis is the ability of *M. tuberculosis* to enter a non-replicating persistent state following colonization of the human lung[12]. Emergence from the persistent state upon immunosuppression, sometimes decades after the initial infection, results in an active infection (reactivation TB) that can be fatal if untreated[13]. The fundamental role of Wbl proteins in developmental processes in Actinobacteria suggested that they could play a role in entry into and emergence from the non-replicative persistent state that is characteristic of *M. tuberculosis* infections. *M. tuberculosis* possesses seven genes encoding Wbl proteins and several of these have been implicated in features of TB pathogenesis such as persistence (WhiB3), antibiotic resistance (WhiB7), and the regulation of lipid and polyketide biosynthesis, including triacylglycerol accumulation as a response to hypoxia and nitrosative stress in macrophages (WhiB3)[14–18]. Transcriptional reprogramming by WhiB3 and WhiB7 is thought to be mediated, at least in part, by interaction with the major sigma factor, $\sigma^A$ [18,19].

The *M. tuberculosis whiB1* gene is essential and encodes a DNA-binding protein with an NO-sensitive [4Fe-4S] cluster[20]. NO is an important component of the host response to *M. tuberculosis* infection; high concentrations of NO generated by activated macrophages can kill *M. tuberculosis* while lower NO levels promote transition to the dormant non-replicating state through activation of the Dos regulon[21,22]. Structure–function studies have shown that all four cysteine residues of WhiB1 are required for iron–sulfur cluster incorporation and that DNA binding by apo- or nitrosylated-WhiB1 requires positively charged residues in the C-terminal region[23].

A greater understanding of the Wbl family of proteins has the potential to open up new opportunities for controlling major bacterial pathogens and enhancing antibiotic production. A major obstacle to realizing this opportunity is the complete lack of three-dimensional structural information for any Wbl protein and direct evidence of interactions with partner proteins. Here the structure of *M. tuberculosis* WhiB1 reveals that it is formed from four $\alpha$-helices, three of which anchor the iron–sulfur cluster. The WhiB1 iron–sulfur cluster is essential for formation of a complex with $\sigma^A$ and a surface of WhiB1 adjacent to the cluster is involved in this interaction. The WhiB1:$\sigma^A$ complex is insensitive to the presence of molecular oxygen ($O_2$) but disassembles upon reaction with NO to release both $\sigma^A$ and WhiB1. Residues implicated in DNA binding by WhiB1 are located in the C-terminal region and the structure suggests that cluster loss (by nitrosylation) disrupts the interaction interface between WhiB1 and $\sigma^A$ permitting the WhiB1 C-terminal helix to bind DNA. The resulting transcriptional reprogramming includes repression of the *espA* operon, which codes for proteins that are essential for the function of the major virulence factor ESX-1.

## Results

**Isolation of a form of WhiB1 with a stable [4Fe-4S] cluster.** *M. tuberculosis* WhiB1 was overproduced in *Mycobacterium smegmatis* with a His-tag and a tobacco etch virus (TEV) protease cleavage site. The protein was isolated as an $O_2$-stable [4Fe-4S]-form as judged by the characteristic absorbance spectrum and an iron content of $3.88 \pm 0.07$ ($n = 3$) atoms per WhiB1 (Supplementary Fig. 1a). The circular dichroism (CD) spectrum of recombinant WhiB1 expressed in *M. smegmatis* closely resembled those of reconstituted WhiB1 and WhiD proteins with positive features at 429 and 512 nm (Supplementary Fig. 1b)[8,20].

**NMR-based structural model of WhiB1.** The observation that the WhiB1 protein remained stable over a period of several days at 25 °C, offered an unprecedented opportunity to apply nuclear magnetic resonance (NMR) techniques to obtain structural information for a Wbl protein. The HSQC NMR spectrum was assigned using standard techniques (Fig. 1a). Of the 89 non-Pro residues following the N-terminal His-tag, all were assigned using standard triple resonance experiments except for residues Val8, Val42, Thr43, Gly61, and Gly62, which could not be observed. A number of other residues relaxed very rapidly and could only be observed using experiments modified for rapidly relaxing signals that are typically found in paramagnetic samples[24,25]; these included Cys9 and Cys37 (Supplementary Fig. 2). The other two cysteine residues, Cys40 and Cys46, also had rapid relaxation, particularly of Cβ. The 1D $^1$H spectra showed broad non-exchangeable signals at 16.6, 15.0, 13.0, 12.4, 12.0, and 10.0 ppm, likely to be due to cysteine Hβ (Supplementary Fig. 3). All these observations are typical of ferredoxin, which in its oxidized form is a [4Fe-4S]$^{2+}$ protein with a diamagnetic ground state but a low energy paramagnetic excited state, such that there is significant unpaired electron density on the iron atoms at room temperature, with consequent paramagnetic broadening of nearby nuclei[24]. Sequence comparisons and modeling websites, such as Phyre, predicted a ferredoxin-like fold at the N-terminal end of WhiB1 and a C-terminal helix[26,27].

NMR was used to generate structural constraints in the form of nuclear overhauser effects (NOEs), chemical shifts, and relaxation rates, which were used to produce a structural model of WhiB1 by restrained molecular dynamics. The model contains a core of three $\alpha$-helices held together by the [4Fe-4S] cluster with a short helical segment between helix1 and helix2 and a fourth C-terminal helix, making a compact structure (Fig. 1b). The chemical shifts of the residues following Ala76 are close to random coil values, implying that the structure becomes disordered beyond this point. Structural comparison using the DALI server produced hits to numerous helical bundles, but nothing of high similarity[28]. Cysteine residues 9, 37, 40, and 46, located in helices 1, 2, and 3, coordinate the [4Fe-4S] cluster and the loop linking helix3 and helix4 that includes Gly61 and Gly62 runs across one face of the cluster (Fig. 1c). These cysteine residues and the $^{58}$GVWGG$^{62}$ motif are characteristic of Wbl proteins. A surface representation of WhiB1 shows that a cluster sulfide atom is exposed at the bottom of a channel whose mouth is formed by Arg10, Phe17, Glu45, and Trp49 (~9 Å diameter)

and likely provides the route for NO to attack the cluster (Supplementary Fig. 4). The C-terminal region of WhiB1 is associated with DNA binding by apo-WhiB1 and presents as a fourth helix (red in Fig. 1b) that lies across the short helical region located between helix1 and helix2 (light blue in Fig. 1b) and the start of helix2 (green in Fig. 1b)[20,23]. The side chains of two residues, Arg73 and Arg74, involved in DNA binding are not accessible to DNA in the holo-WhiB1 structure[23]. Therefore, the structure suggests that cluster disassembly (by nitrosylation or by removal of iron) frees the C-terminal helix permitting DNA binding. Far UV CD spectroscopy showed that the spectrum of holo-WhiB1 was characteristic of an α-helical protein (with double minima at 207 and 219 nm and a crossover point at 199 nm), whereas that of apo-WhiB1 lost most of the helical structure (minimum at 203 nm with a negative signal at 198 nm) indicative of conformational changes that could modulate DNA-binding activity (Supplementary Fig. 5a). In agreement with this, the NMR spectrum of apo-WhiB1, prepared by nitrosylation and re-isolation, was very different from that of the holo-protein (Supplementary Fig. 5b). Most of the signals from the holo-protein moved into more random coil positions, with ${}^1$H chemical shifts in the range 7.5–8.5 ppm. There were ~15 signals missing, typical of a molten globule structure, and consistent with a proportion of folded structure in equilibrium with random coil. Interestingly, the small number of signals that remained in the same position in the apo- and holo-protein include many residues in the unstructured N and C termini, but also residues 69, 73, 75, and 78, which all lie on the outer face of the C-terminal helix, implying that the only part of the protein that retains a native-like folded structure in the apo-protein is the C-terminal helix. This observation is consistent with the suggestion that loss of the cluster liberates the C-terminal helix to permit binding to DNA.

**The WhiB1 [4Fe-4S] cluster reacts with NO.** Electrospray ionization mass spectrometry (ESI-MS) under non-denaturing conditions was used to analyze purified WhiB1[29]. The $m/z$ spectrum in the range 500–3000 $m/z$ is shown in Supplementary Fig. 6a. The deconvoluted spectrum revealed a major species at 11,986 Da due to monomeric [4Fe-4S] WhiB1 (predicted mass 11,986 Da; Fig. 2a and Supplementary Fig. 6b). Thus, ESI-MS provides additional evidence, alongside spectroscopic data, that Wbl proteins possess [4Fe-4S] clusters[8–10,16,20]. Several low abundance (≤20%) adducts were detected at higher masses; the clearest of which corresponds to an oxygen adduct (+16 Da). In the lower mass region, a minor peak at 11,632 Da was assigned as apo-WhiB1 with two disulfide bonds (measured mass is 4 Da lower than the predicted WhiB1 mass). Additional peaks at 11,664 and 11,696 Da corresponded to apo-WhiB1 with one and two sulfur adducts (+32 and +64 Da), respectively. A further peak at 11,953 Da is most likely due to a small amount of WhiB1 containing a [4Fe-3S] cluster resulting from loss of sulfide.

In-source collision-induced dissociation (isCID) of holo-WhiB1 resulted in a complex series of overlapping peaks, which were subdivided into two distinct groups corresponding to protein bound cluster fragments (11,780–11,970 Da) and sulfur adducts of apo-WhiB1 (11,600–11,710 Da). Major protein bound cluster fragments were observed at 11,953, 11,919, 11,884, and 11,851 Da, corresponding to [4Fe-3S], [4Fe-2S], [3Fe-2S]-Na, and [3Fe-S]-Na forms of WhiB1, respectively (Supplementary Table 1). In general, these cluster peaks at lower mass values represent the loss of inorganic sulfide (S) or iron-sulfide (Fe-S) from the protein bound species. The sequential loss of inorganic sulfide has been observed previously for [3Fe-4S] ferredoxins[30]. The lack of [3Fe-4S] or [3Fe-3S] cluster species is consistent with

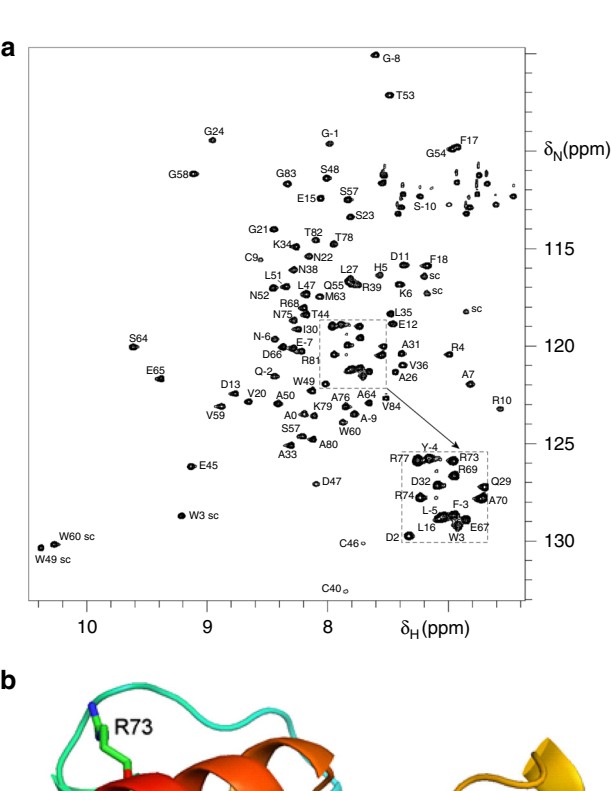

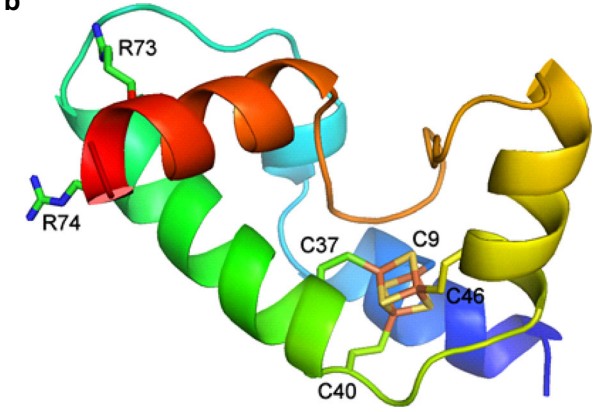

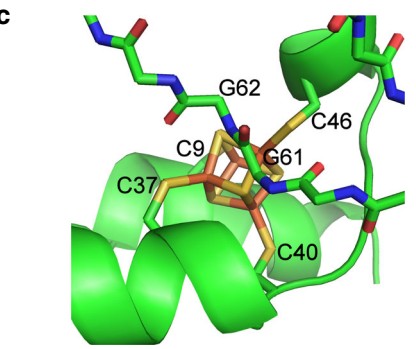

**Fig. 1** NMR structural model. **a** HSQC NMR spectrum of WhiB1. Note the weak intensity and unusual chemical shifts for Cys66 and Cys60. Residues in the TEV tag have negative residue numbers. sc, side chain. **b** Structural model of WhiB1 calculated using NMR restraints. The chain is colored from blue at the N terminus to red at the C terminus. The structure shown starts at residue 1, and finishes at residue 84. The two arginine residues 73 and 74 in the C-terminal helix are predicted to interact with DNA. For clarity amino acid residues are indicated using the single letter code. **c** The environment of the WhiB1[4Fe-4S] cluster. The only side chains shown are the coordinating cysteine residues. The protein is displayed as a cartoon, except for the loop containing Gly61 and Gly62, which is shown as stick representations

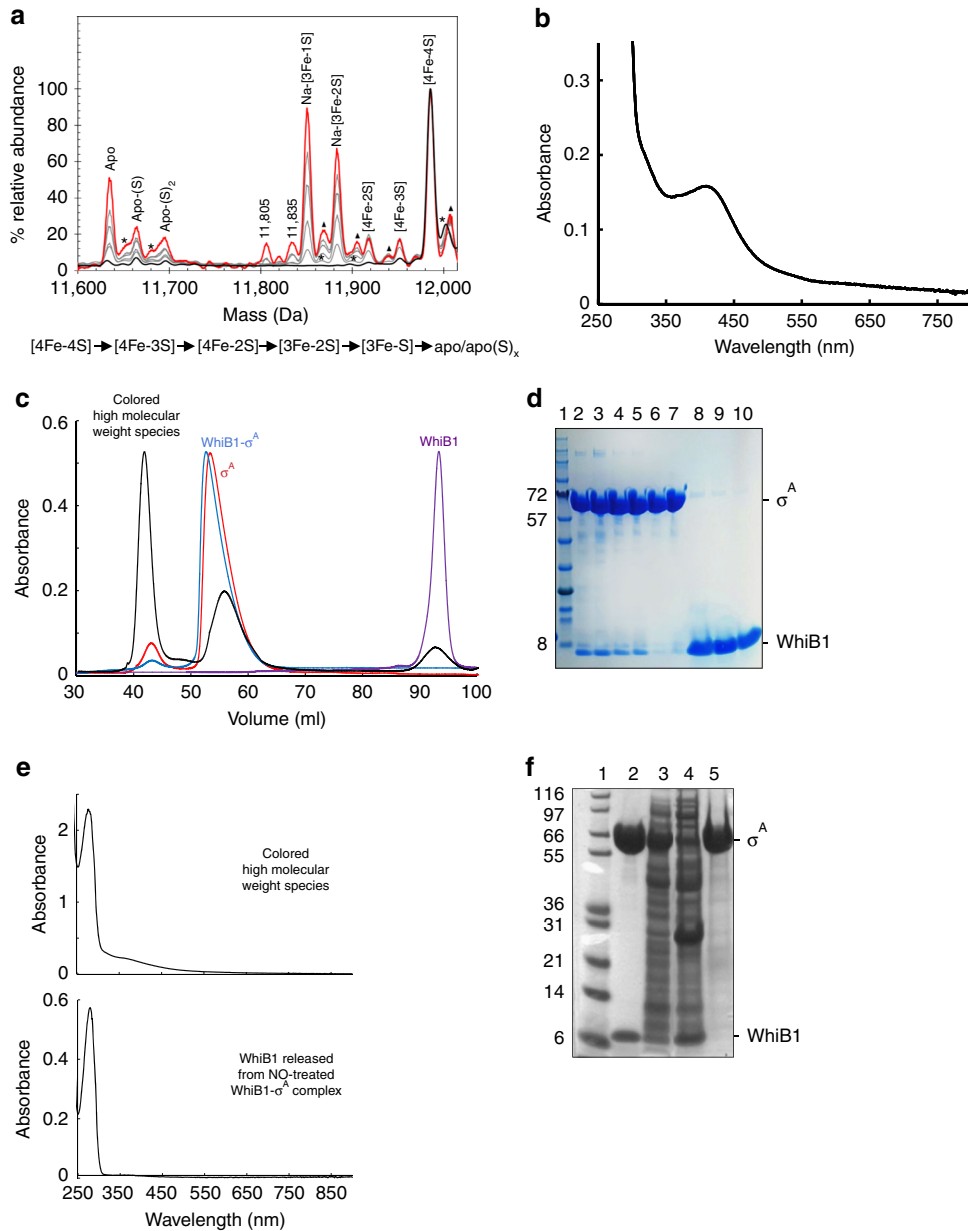

**Fig. 2** In-source collision-induced dissociation of WhiB1 and isolation of a holo-WhiB1:σ$^A$ complex. **a** Deconvoluted ESI-MS spectra of [4Fe-4S] WhiB1 (black spectrum). The [4Fe-4S] cluster bound form is the only significant species. Application of isCID with increasing energy (up to 140 eV, red spectrum, intermediate energies gave spectra in gray) resulted in detection of cluster fragment species, along with apo-WhiB1 and sulfur adducts, as indicated. Asterisks indicate oxygen adducts and triangles indicate sodium adducts (in addition to those indicated as major species). Note that sodium adducts increase with isCID. Peaks annotated with mass numbers could not be unambiguously assigned. WhiB1 (20 μM [4Fe-4S]) was in 250 mM ammonium acetate, pH 8.0. A reaction scheme based on the mass spectrometry data is shown below the spectra. **b** Absorbance spectrum of the protein complex isolated from cell extracts of *E. coli* transformed with plasmids expressing His-tagged *M. tuberculosis* σ$^A$ (His-σ$^A$) and untagged WhiB1 by nickel affinity chromatography. **c** Gel filtration elution profiles (280 nm) of His-σ$^A$ (red trace), WhiB1 (purple trace), His-σ$^A$-WhiB1 (blue trace), and His-σ$^A$-WhiB1 after exposure to NO (black trace). **d** SDS-PAGE analysis of the fractionated NO-treated His-σ$^A$-WhiB1 (black trace in **c**) as follows: lane 1, protein molecular weight markers; lane 2, nitrosylated His-σ$^A$-WhiB1 sample applied to the column; lanes 3–5, species eluting ~42 ml; lanes 6 and 7, species eluting at ~56 ml; lanes 8–10, species eluting at ~93 ml. **e** UV-visible spectrum of fractions eluting at ~42 ml (upper panel) and fractions eluting at ~93 ml (lower panel). **f** SDS-PAGE analysis of WhiB1:His-σ$^A$ complexes. Lane 1, molecular weight markers; lane 2, isolated WhiB1:His-σ$^A$; lane 3, cell-free extract of *E. coli* expressing His-σ$^A$ and WhiB1(Cys40Ala); lane 4, flow through from nickel affinity column after application of the cell-free extract shown in lane 3; lane 5, eluate from the nickel affinity column loaded with the cell-free extract shown in lane 3. The sizes of the standard proteins and the locations of WhiB1 proteins and His-σ$^A$ (σ$^A$) are indicated

solution studies of WhiB1 in which little or no EPR active species were observed following cluster disassembly due to O$_2$ exposure[20]. The data indicate that the degradation of the WhiB1 [4Fe-4S] cluster is initiated by the loss of a sulfur atom, consistent with an exposed cluster sulfide (Supplementary Fig. 4), followed by

ejection of further sulfur and iron atoms to yield apo- and persulfide-forms of WhiB1 (Fig. 2a).

The chemical promiscuity of sulfur means that the precise identity of the persulfide-modified cysteine residues is unknown, but sulfane sulfur stored in this way could be significant in

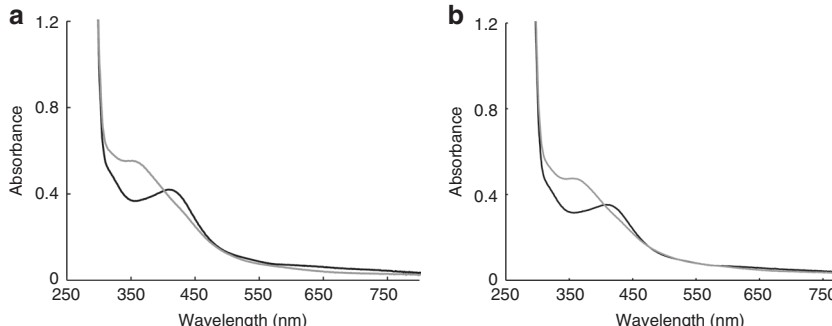

**Fig. 3** Reaction of the iron–sulfur cluster of the His-σ$^A$-WhiB1 complex with NO. **a** Absorbance spectrum of isolated WhiB1:His-σ$^A$ before (black line) and after (gray line) treatment with 14-fold molar excess NO. **b** Absorbance spectrum of WhiB1:His-σ$^A$CTD before (black line) and after (gray line) treatment with 15-fold molar excess NO

repairing the WhiB1 [4Fe-4S] cluster[31]. A similar pattern of cluster disassembly was observed during ESI-MS isCID experiments with the WhiD protein from *S. coelicolor* (Supplementary Fig. 7), suggesting that all Wbl protein [4Fe-4S] clusters behave similarly. As previously shown for reconstituted WhiB1 expressed in *E. coli*, the iron–sulfur cluster of WhiB1 reacted with NO (Supplementary Fig. 8a)[8,20]. Titration with NO indicated that no further changes in the absorbance spectrum occurred at [NO]: [4Fe-4S] ≥8 (Supplementary Fig. 8b). Similar titrations of reconstituted *S. coelicolor* WhiD and WhiB1 expressed in *E. coli*, as well as the unrelated *E. coli* transcriptional regulator FNR, under anaerobic conditions indicated that ~8 NO molecules reacted with each iron–sulfur cluster[8,20,32]. Thus, the previous conclusion, based on observations with reconstituted protein, that *M. tuberculosis* WhiB1 possesses an NO-sensitive, O$_2$-stable [4Fe-4S] cluster, are supported by the analysis of the un-reconstituted protein overproduced in the mycobacterium *M. smegmatis*.

**Holo-WhiB1 interacts with *M. tuberculosis* σ$^A$.** A survey of the monomeric transcription regulators of *M. tuberculosis* showed that WhiB1 interacted with the major sigma factor, σ$^A$[33]. Aerobic purification of an N-terminally His-tagged *M. tuberculosis* σ$^A$ (His-σ$^A$) co-expressed in *E. coli* with an untagged WhiB1 resulted in the isolation of a colored complex with an absorbance spectrum (broad absorbance located around 420 nm) indicative of the presence of a [4Fe-4S] cluster (Fig. 2b). Size exclusion chromatography under aerobic conditions showed the presence of a single major species (Fig. 2c, blue trace) that was composed of His-σ$^A$ and [4Fe-4S]-WhiB1 (Fig. 2f, lane 2). Moreover, after purification, the in vivo assembled WhiB1:His-σ$^A$ complex was stable under aerobic conditions for >2 weeks, i.e., the WhiB1 [4Fe-4S] cluster and the interaction with σ$^A$ remained intact during this time, indicating that the WhiB1 iron–sulfur cluster is unlikely to act as an O$_2$ sensor. When *sigA* was co-expressed with *whiB1* coding for a variant, WhiB1(Cys40Ala), that cannot acquire a [4Fe-4S] cluster[23], a WhiB1:His-σ$^A$ was not formed, indicating that the iron–sulfur cluster is required for interaction with σ$^A$ (Fig. 2f, lanes 3–5).

**NO releases apo-WhiB1 from the WhiB1:His-σ$^A$ complex.** The observation that the WhiB1 iron–sulfur cluster was required for interaction with σ$^A$ (Fig. 2f) suggested that reaction of the cluster with NO could initiate breakdown of the WhiB1:σ$^A$ complex, liberating the DNA-binding form (apo/nitrosylated) of WhiB1[20]. The absorbance spectrum obtained after addition of NO to the complex (Fig. 3a) was similar to that observed for the [4Fe-4S] form of WhiB1 after NO treatment (Supplementary Fig. 8a).

Titration of the WhiB1:His-σ$^A$ complex with NO showed that the iron–sulfur cluster reacted with ~8 NO molecules, as observed for the isolated WhiB1 protein (Supplementary Fig. 9). Size exclusion chromatography after treatment of the holo-WhiB1:σ$^A$ complex with NO and analysis of the fractions by denaturing polyacrylamide gel electrophoresis (SDS-PAGE) and absorbance spectroscopy revealed the presence of liberated σ$^A$ and apo-WhiB1, as well as a high molecular mass species composed of WhiB1 and σ$^A$ with an absorbance spectrum consistent with the presence of a nitrosylated iron–sulfur cluster (shoulder at ~360 nm) (Fig. 2c, black trace; Fig. 2d, e). This suggested that the reaction of WhiB1:His-σ$^A$ with NO resulted in formation of a nitrosylated complex that disassembles to yield apo-WhiB1 and σ$^A$.

**The interface between WhiB1 and σ$^A$.** *M. tuberculosis* σ$^A$ is a member of the σ$^{70}$ family of sigma factors, which are characterized by five structural domains[34]. The C-terminal domain contains conserved regions 4.1 and 4.2 that are involved in recognition of the core −35 element of σ$^{70}$-dependent promoters and in interactions with transcription regulators. Both *M. tuberculosis* WhiB3 and WhiB7 have been shown to act as redox-sensitive transcriptional regulators by binding to region 4 of σ$^A$[18,19]. To determine whether this was also true for WhiB1, a His-tagged version of the C-terminal domain (amino acids Ala447–Asp528) of *M. tuberculosis* σ$^A$ (His-σ$^A$CTD) was co-expressed with *whiB1* in *E. coli*. Purification of His-σ$^A$CTD using affinity chromatography yielded a colored complex consisting of holo-WhiB1 and His-σ$^A$CTD, indicating that, like WhiB3 and WhiB7, WhiB1 interacts with region 4 of σ$^A$ (Fig. 3b). As was the case for the WhiB1:σ$^A$ complex, the iron–sulfur cluster of the WhiB1:σ$^A$CTD complex was stable under aerobic conditions (Supplementary Fig. 10) but was reactive with NO (Fig. 3b).

Changes in the chemical shifts observed in two-dimensional ($^1$H and $^{15}$N) NMR spectra provide a means to define regions of interaction between proteins. Comparison of NH weighted chemical shift changes between WhiB1 and the WhiB1:His-σ$^A$CTD complex indicated that the environment of the C-terminal region of WhiB1, which is implicated in DNA binding, was similar in both (Fig. 4a). This suggests that σ$^A$ does not interact with the DNA-binding region of WhiB1. The largest weighted NH chemical shift changes were for His5, Glu12, Gly54, Ser57, and Gly58, suggesting that the region adjacent to the iron–sulfur cluster forms a surface that participates in interactions with σ$^A$CTD (Fig. 4b). This is consistent with the observation that the iron–sulfur cluster is essential for formation of the WhiB1:σ$^A$ complex (Fig. 2f).

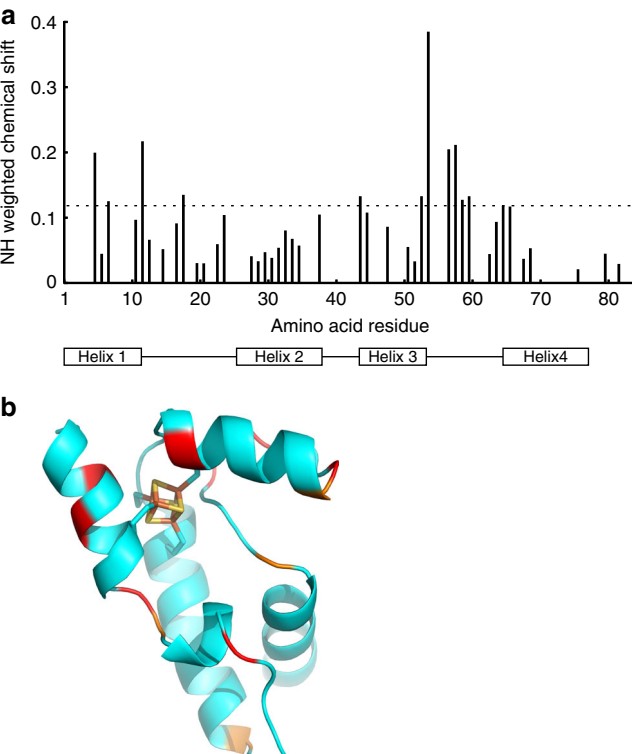

**a**

**b**

**Fig. 4** Weighted NH chemical shift differences for WhiB1 compared to the WhiB1:His-σ$^A$CTD complex. **a** The chart shows the weighted ($d = \sqrt{\frac{1}{2}(\delta_H)^2 + (\alpha^*\delta_N^2)}$) chemical shifts for the WhiB1:His-σ$^A$CTD complex compared to the WhiB1 protein. The value of $\alpha$ was 0.14 except for glycine residues, where a value of 0.2 was used[56]. The dashed line marks the average chemical shift plus one standard deviation. The rectangles below the chart show the four (1–4) α-helices of WhiB1. **b** Shifts are mapped onto the structure of WhiB1. The largest shift changes (red) are in the N-terminal region and the linker between helix3 and helix4. The C-terminal DNA-binding helix (bottom right) shows very little shift change

**Reaction of the WhiB1:σ$^A$CTD complex with NO.** Native ESI-MS of the WhiB1:σ$^A$CTD complex revealed a major species at 21,191 Da, corresponding to a 1:1 complex of [4Fe-4S] WhiB1 and σ$^A$CTD, with less abundant peaks at 9668 and 11,522 Da, corresponding to free [4Fe-4S] WhiB1 and σ$^A$CTD, respectively (Supplementary Fig. 11a, b and Supplementary Table 2); optimization of conditions for complex detection reduced the intensity of the latter peaks (Supplementary Fig. 11c).

Incremental increases in the isCID voltage resulted in progressive loss of the WhiB1:σ$^A$CTD complex (21,191 Da) and the appearance of two peaks corresponding to complexes with [4Fe-3S] and [4Fe-2S] clusters (32 and 64 Da less mass) (Supplementary Fig. 11d). The low abundance of the [4Fe-3S] and [4Fe-2S] forms is consistent with cluster damage initiating dissociation of the WhiB1:σ$^A$CTD complex. In the dissociated WhiB1 mass region, the [4Fe-4S] species was observed to increase and then decrease (Supplementary Fig. 11e). The peak due to [4Fe-4S] WhiB1 increased steadily (as WhiB1 dissociated from the complex) with increasing collision energy, but at energies above ~100 eV it decreased again as the cluster itself began to fragment, resulting in the same cluster breakdown products as observed for [4Fe-4s] WhiB1 alone (Fig. 2a).

Size exclusion chromatography suggested that exposure to NO caused the disassembly of the WhiB1:σ$^A$ complex (Fig. 2c). ESI-MS analysis of the reaction of the WhiB1:σ$^A$CTD complex with NO showed that the complex peak (21,191 Da) decreased in

intensity as the ratio of NO to [4Fe-4S] increased (Fig. 5a). This was accompanied by increased intensities of peaks corresponding to σ$^A$CTD (11,522 Da) and apo-WhiB1 (9315 Da) (Fig. 5b, c). In addition to the latter, single (apo-WhiB1(S)) and double (apo-WhiB1(S)$_2$) persulfide species were also observed, indicating that some of the cluster sulfide was oxidized to S$^0$ (sulfane), becoming covalently attached to WhiB1 during the nitrosylation reaction (Fig. 5c). This has been observed previously for other iron–sulfur cluster reactions[32]. Plots of relative abundance as a function of the NO:[4Fe-4S] ratio showed that the loss of the cluster/dissociation of the complex into σ$^A$CTD and apo-WhiB1 species occurs linearly, with reaction complete at ~8 NO per cluster (Fig. 5d–h), consistent with the gel filtration and titration data reported above and previous studies of Wbl proteins[8,20]. This demonstrates that the nitrosylation reaction occurs in a concerted manner, such that the reaction of NO with one [4Fe-4S] cluster goes to completion before a second cluster undergoes reaction. Consequently, at ratios below 8 NO per cluster unaffected WhiB1:σ$^A$CTD complexes remain intact. This may account for the absence of cluster degradation/nitrosylation species (e.g., adducts of NO) in the ESI-MS data, because intermediate species do not accumulate in a concerted reaction and hence only the end product apo-WhiB1 species are readily detected following reaction with NO. Because some species ionize better than others ESI-MS is not quantitative and therefore there might be additional products that are not detected, which could include the rapidly eluting nitrosylated complex apparent in gel filtration experiments (Fig. 2c).

**The WhiB1:σ$^A$ complex is disrupted by exposure to NO in vivo.** Interaction of WhiB1 and σ$^A$CTD was assessed using a bacterial two-hybrid approach[35]. Significant interaction, as indicated by β-galactosidase activities ~20-fold greater than the negative control (E. coli JRG5386) and ~threefold lower than the positive control (E. coli JRG5387), was detected in cultures of E. coli JRG2862, which expressed the T25-WhiB1 and T18-σ$^A$CTD fusions (Fig. 5i). In vitro analyses (Fig. 2f) indicated that the WhiB1 iron–sulfur cluster was required for formation of a complex with σ$^A$. Restricting the availability of iron by supplementing culture medium with dipyridyl decreased β-galactosidase activity in the E. coli JRG6862 cultures, indicating that interaction of WhiB1 and σ$^A$CTD was impaired, consistent with the in vitro data. In contrast, the output from E. coli JRG5387, the leucine zipper control, was unaffected by iron starvation (Fig. 5i). A major host response to bacterial infection, including the human response to an M. tuberculosis infection, is to restrict the access of the bacteria to sources of iron[36,37]. This suggests that the WhiB1:σ$^A$ interaction could be disassembled in response to host-mediated iron starvation.

The effect of NO on formation of the WhiB1:σ$^A$ complex was tested in anaerobic cultures grown in the presence of nitrite at pH 5.5. A significant reduction in β-galactosidase activity was observed compared to the control cultures (Fig. 5i). Thus, despite the presence of the endogenous NO sensing and detoxification systems of E. coli, the recombinant WhiB1 was able to respond, consistent with a role as an NO-responsive regulator in M. tuberculosis. There was no significant difference in complex formation when aerobic cultures were compared to those grown under anaerobic (fumarate respiratory) conditions, suggesting that the WhiB1 iron–sulfur cluster was stable in the presence of O$_2$ in vivo (Fig. 5i). Despite the use of an heterologous host, the demonstration of disassembly of the WhiB1:σ$^A$ complex in vivo under conditions of iron starvation and in the presence of NO, but not in response to O$_2$, is consistent with the in vitro analyses of the WhiB1:σ$^A$ complex and suggests a role for WhiB1 as an NO- and iron-responsive sensor regulator during M. tuberculosis infections. Thus, the NO-mediated liberation of σ$^A$ and DNA-

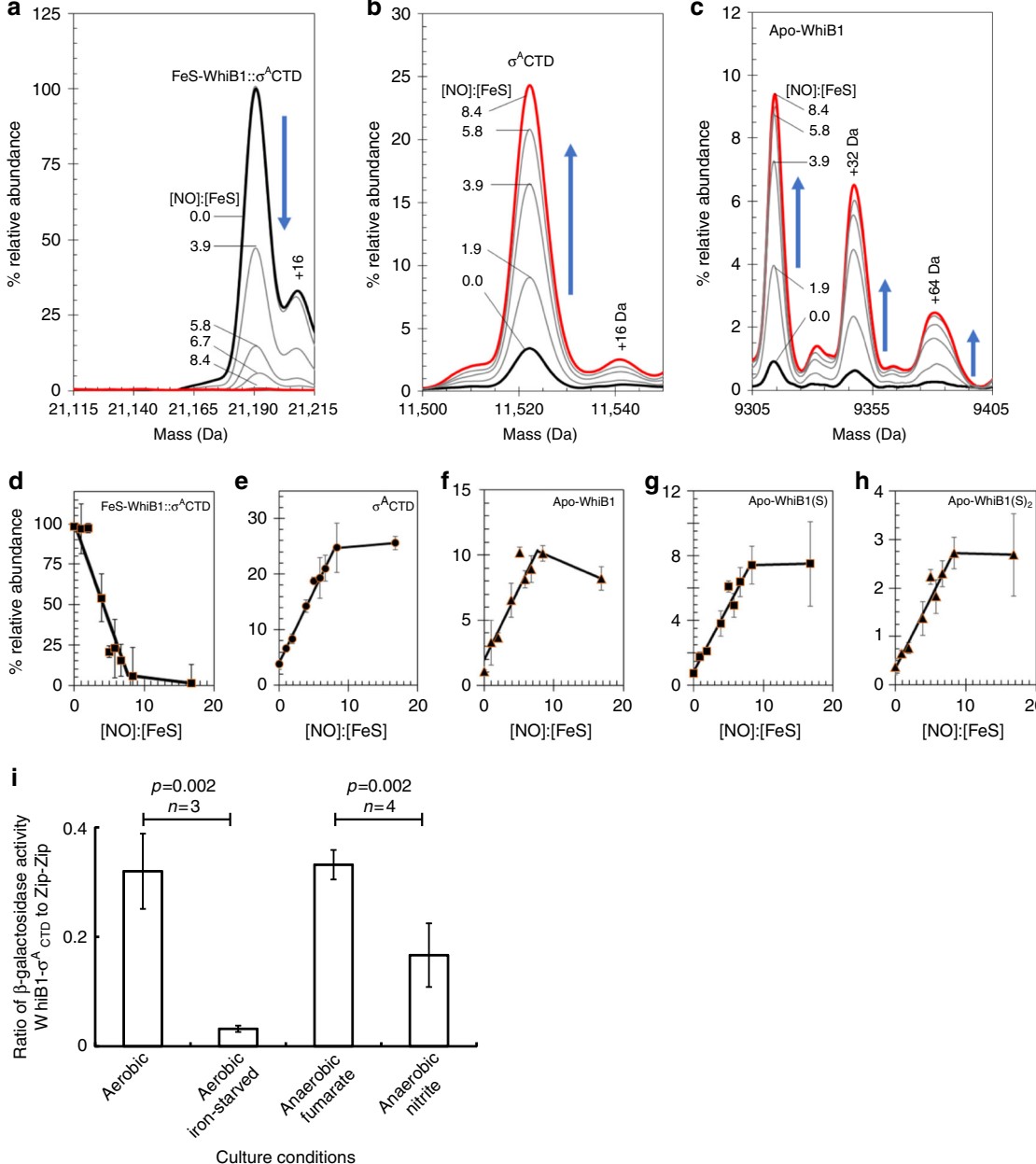

**Fig. 5** The effect of NO on the WhiB1:$\sigma^A$CTD complex. **a–c** Deconvoluted mass spectra in the WhiB1:$\sigma^A$CTD complex (**a**), $\sigma^A$CTD (**b**), and apo-WhiB1 (**c**) mass regions as a function of increasing ratios of NO to [4Fe-4S] WhiB1. **d–h** Plots of relative abundance of WhiB1:$\sigma^A$CTD complex (**d**), $\sigma^A$CTD (**e**), apo-WhiB1 (**f**), apo-WhiB1 with one additional sulfur (apo-WhiB1(S)) (**g**) and apo-WhiB1 with two additional sulfur adducts (apo-WhiB1(S)$_2$) (**h**) as a function of the NO:[4Fe-4S] ratio. WhiB1:$\sigma^A$CTD complex (11 µM [4Fe-4S]) was in 250 mM ammonium acetate, pH 8.0. The error bars represent the standard deviations for average mass spectra data ($n = 6$). **i** Bacterial two-hybrid analysis of interaction between *M. tuberculosis* WhiB1 and $\sigma^A$CTD fused to the T25 and T18 domains, respectively, of *Bordetella pertussis* adenylate cyclase in *E. coli* BTH101. Interaction between the fusion partners is manifest as enhanced production of β-galactosidase. Control cultures were of the same host transformed with plasmids encoding fusions to the GCN4 leucine zipper (Zip-Zip)[35]. The data are shown as the β-galactosidase activities obtained for WhiB1 and $\sigma^A$CTD divided by those obtained for control (Zip-Zip) cultures grown under the indicated conditions. The chart shows the mean and standard deviation for independent cultures; significance (*t* test) is indicated

binding competent conformers of WhiB1 is likely to initiate changes in gene expression in *M. tuberculosis* to a key component of the host immune response.

**Underexpression of *whiB1* dysregulates ESX-1.** *M. tuberculosis* *whiB1* is an essential gene and it was not possible to create a *whiB1* deletion mutant in the absence of a complementing plasmid[20]. The level of expression of *whiB1* in the complemented mutant was affected by how much flanking DNA was present in

the construct, there being decreasing levels of *whiB1* expression with decreasing size of the construct (DMH3 > DMH2 > DMH1; Supplementary Table 3 and Supplementary Fig. 12). Transcript profiling of the partially complemented *whiB1* mutant with the lowest amount of *whiB1* expression (DMH1) compared to the parent carrying the same complementing plasmid showed that 35 genes (25 operons) exhibited altered expression (>threefold; *p* < 0.005) (Table 1). Expression of *whiB1* was 2.7-fold lower in the complemented mutant. Among the genes deemed to exhibit altered expression, 12 coded for hypothetical proteins, 9 (*lipQ*,

**Table 1 Genes with altered expression when *whiB1* was underexpressed**

| Gene | Rv number | Fold change | Function | Operon |
|------|-----------|-------------|----------|--------|
| *espD* | Rv3614c | 20.9 | ESX-1 secretion-associated protein EspD | *espA-E* |
| *espC* | Rv3615c | 19.8 | ESX-1 secretion-associated protein EspC | *espA-E* |
| *espA* | Rv3616c | 13.1 | ESX-1 secretion-associated protein EspA | *espA-E* |
| *Rv3613c* | Rv3613c | 11.5 | Hypothetical protein | *espA-E* |
| *Rv2660c* | Rv2660c | 6.1 | Hypothetical protein | *Rv2660c* |
| *Rv3662c* | Rv3662c | 5.8 | Hypothetical protein | *Rv3662c* |
| *papA1* | Rv3824c | 5.3 | Acyltransferase | *pks2-papA1-mmpL8* |
| *mftE* | Rv0695 | 5.2 | Mycofactocin system creatinine amidohydrolase family protein | *mftEF* |
| *mmpL8* | Rv3823c | 5.0 | Integral membrane transport protein MmpL8 | *pks2-papA1-mmpL8* |
| *mftF* | Rv0696 | 4.7 | Mycofactocin biosynthesis glycosyltransferase | *mftEF* |
| *Rv3612c* | Rv3612c | 4.5 | Hypothetical protein | *espA-E* |
| *ppsE* | Rv2935 | 4.4 | Phthiocerol synthesis polyketide synthase type I | *ppsE* |
| *Rv0755A* | Rv0755A | 4.2 | Transposase | *Rv0755A* |
| *Rv2159c* | Rv2159c | 4.1 | Hypothetical protein | *Rv2159c* |
| *Rv1638A* | Rv1638A | 4.1 | Hypothetical protein | *Rv1639c-Rv1638A* |
| *lipX* | Rv1169c | 4.1 | Lipase | *lipX* |
| *Rv1986* | Rv1986 | 4.1 | Amino acid transporter | *Rv1986* |
| *Rv3572* | Rv3572 | 4.0 | Hypothetical protein | *Rv3572* |
| *pks2* | Rv3825c | 3.8 | Phthioceranic/hydroxyphthioceranic acid synthase | *pks2-papA1-mmpL8* |
| *tgs2* | Rv3734c | 3.5 | Diacyglycerol O-acyltransferase | *tgs2* |
| *Rv2632c* | Rv2632c | 3.5 | Hypothetical protein | *Rv2633c-Rv2632c* |
| *Rv1639c* | Rv1639c | 3.4 | Hypothetical protein | *Rv1639c-Rv1638A* |
| *Rv0108c* | Rv0108c | 3.3 | Hypothetical protein | *Rv0108c* |
| *lat* | Rv3290c | 3.3 | L-lysine-epsilon aminotransferase | *lat* |
| *Rv0888* | Rv0888 | 3.3 | Hypothetical protein | *Rv0888* |
| *pks3* | Rv1180 | 3.3 | Polyketide beta-ketoacyl synthase | *pks3* |
| *Rv2633c* | Rv2633c | 3.2 | Hypothetical protein | *Rv2633c-Rv2632c* |
| *Rv2459* | Rv2459 | 3.2 | MFS-type transporter | *Rv2459* |
| *lipQ* | Rv2485c | 3.1 | Carboxylesterase LipQ | *lipQ* |
| *Rv3633* | Rv3633 | 3.0 | Hypothetical protein | *Rv3633* |
| *Rv1393c* | Rv1393c | 3.0 | Monoxygenase | *Rv1393c* |
| *mbtJ* | Rv2385 | 0.3 | Acetyl hydrolase | *mbtJ* |
| *cyp121* | Rv2276 | 0.3 | Cytochrome P450 | *cyp121* |
| *mbtI* | Rv2386c | 0.3 | Salicylate synthase | *mbtI* |

*lipX*, *mbtI*, *mbtJ*, *papA1*, *ppsE*, *pks2*, *pks3*, and *tgs2*) have functions in lipid metabolism, 3 (*mmpL8*, *Rv1986*, and *Rv2459*) code for transporters, 5 (*cyp121*, *lat*, *mftE*, *mftF*, and *Rv1393c*) have functions in intermediary metabolism and respiration, and 1 (*Rv0755A*) code for a transposase. The presence of both up- (22 operons) and downregulated (3 operons) genes suggests that WhiB1 can act (directly or indirectly) as both a repressor and an activator of gene expression.

The genes with the largest changes in expression were those of the *espA* operon, which codes for proteins required for ESX-1 secretion activity[38–40]. ESX-1 secretes the potent antigens ESAT-6 (EsxA) and CFP-10 (EsxB), as well EspA and EspC, to cause phagosome dysfunction permitting dissemination of *M. tuberculosis* from the phagosome to the cytoplasm and is therefore a major factor in tuberculosis pathogenesis[41–44]. Because of the mutually dependent nature of the ESX-1 substrates, ESX-1 activity is controlled by expression of the *espA* operon[38]. Regulation of the *espA* operon by WhiB1 was confirmed by qRT-PCR of the partially complemented *whiB1* mutants, which showed that expression of *espA* negatively correlated to *whiB1* expression, i.e., there was more *espA* expression with less expression of *whiB1* (Supplementary Fig. 12). This regulation is likely to be direct because apo-WhiB1, prepared using the *E. coli* expression system, bound specifically to the *espA* promoter region (Supplementary Figs. 13–15).

## Discussion

The isolation of *M. tuberculosis* WhiB1 with a [4Fe-4S] cluster that was stable for several weeks in air permitted the determination of the structure of a Wbl protein. Wbl proteins have been extensively studied since their discovery more than 40 years ago because of their roles in regulating fundamental aspects of actinobacterial biology, including developmental processes and virulence. The WhiB1 structure provides a framework for better understanding the roles of Wbl protein iron–sulfur clusters in signal perception and in mediating interaction with $\sigma^A$, as well as how the C-terminal helix is released to allow binding to DNA. The application of mass spectrometry to investigate the NO-mediated disassembly of the holo-WhiB1:$\sigma^A$ complex demonstrates the power of this technique in providing mechanistic understanding of the response of complex regulatory systems to their cognate signals. NO-triggered disassembly of the WhiB1:$\sigma^A$ complex is likely during *M. tuberculosis* infections, where NO production by lung macrophages is a major determinant of the outcome of an infection[21,22]. The NO-dependent disassembly of the WhiB1:$\sigma^A$ complex and determining the *M. tuberculosis* WhiB1 regulon by underexpressing *whiB1* provides further insight into the roles played by Wbl proteins in TB pathogenesis (Fig. 6). Notably, WhiB1-mediated transcriptional reprogramming included of components of the ESX-1 secretion system, which is the major *M. tuberculosis* virulence factor, being involved in invasion of host cells, phagosome escape, bacterial dispersal, and forming extensive tissue lesions (Fig. 6)[45]. Interestingly, in *Mycobacterium marinum* another Wbl protein, WhiB6, acts as a dual regulator ESX-1 depending on the state of its iron–sulfur cluster[46]. The newly identified regulation of ESX-1 function by WhiB1 should prompt further research into how Wbl proteins work together to coordinate virulence gene expression in Mycobacteria through interactions with DNA and partner proteins.

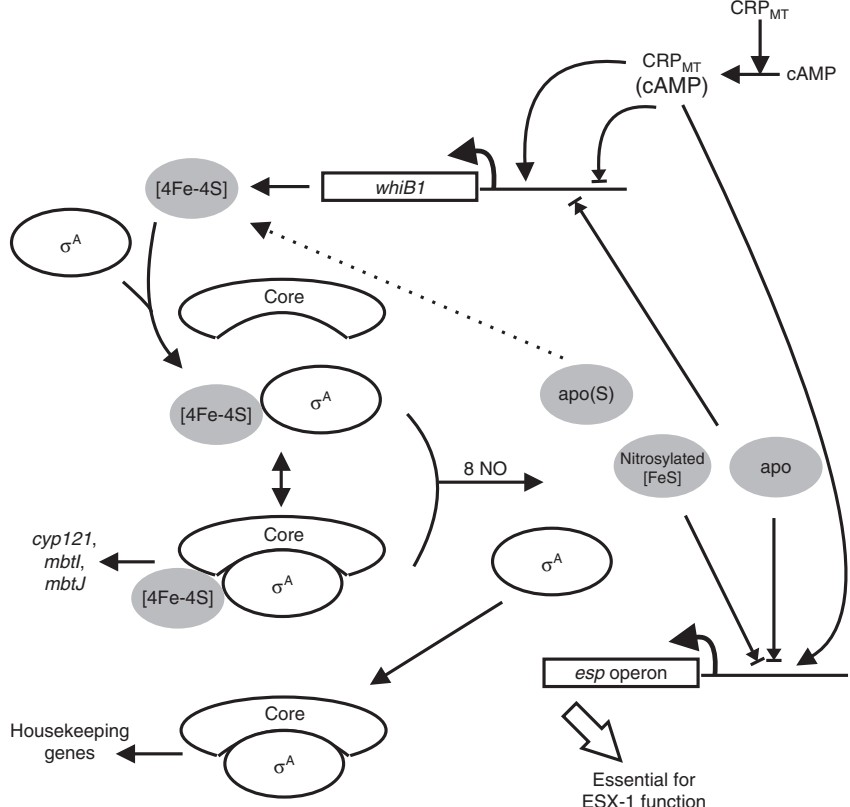

**Fig. 6** *Mycobacterium tuberculosis* WhiB1 as a NO-responsive regulator of gene expression. Expression of *whiB1* is controlled by the cyclic AMP receptor protein (CRP) in response to cAMP (dual regulation) and by apo-WhiB1 (negative regulation)[20,59,60]. In the absence of NO, holo-WhiB1 (gray ellipse, [4Fe-4S]) forms a complex with the major sigma factor (open ellipse, $\sigma^A$). The WhiB1:$\sigma^A$ complex is likely to be capable of interacting with core RNA polymerase (core) to activate a subset of *M. tuberculosis* genes, including *cyp121*, *mbtI*, and *mbtJ*, as indicated by the gene expression profiling of cultures underexpressing *whiB1* (Table 1). When *M. tuberculosis* is exposed to NO, the WhiB1 iron–sulfur cluster is nitrosylated, reacting with 8 NO molecules. This results in the liberation of DNA-binding forms of WhiB1 (gray ellipse, apo, and nitrosylated [FeS]) leading to the repression of multiple genes, including the *espA* operon, which is essential for the function of the virulence-critical ESX-1 secretion system and is also regulated by CRP[61]. Thus, CRP and WhiB1 combine to integrate inputs from two signaling molecules (cAMP and NO) associated with infection at the *esp* operon promoter. Intoxication of macrophages by *M. tuberculosis* derived cAMP is suggested to promote growth and would via the action of the CRP(cAMP) complex activate *esp* operon expression and ESX-1-mediated secretion[62]. Precise control of ESX-1 is required because although the secreted effector proteins are essential for infection they are highly antigenic; NO could act as a signal indicating immune system activity resulting in a WhiB1-mediated repression of *esp* operon expression and shutdown of ESX-1 activity (Supplementary Fig. 13). The retention of sulfur (as persulfides) by WhiB1 offers a route to cluster repair without the need for cysteine desulfurase when NO has been detoxified (dashed line)[31]

## Methods

**Microbiological methods**. Bacterial strains and plasmids and their sources are listed in Supplementary Table 3. Luria–Bertani (LB) medium (Sigma-Aldrich) was used for growing *E. coli* cultures at 37 °C with shaking at 250 rpm. DNA was handled using standard methodologies[47]. For overproduction of recombinant WhiB1 in *E. coli* BL21 (λDE3) pGS2500 was created by amplifying the *whiB1* open reading frame from *M. tuberculosis* H37Rv genomic DNA using ECWb1F and ECWb1R primers (Supplementary Table 4) and ligating the product, after digestion with NdeI and SalI into pET28a. The plasmid pGS2500 coded for WhiB1 (His$_6$-Thrombin cleavage site-WhiB1). For isolation, a non-tagged WhiB1 from *E. coli*, the expression plasmid (pGS2560), was created using the same amplified DNA but ligation into the pET21a vector.

*Mycobacterium smegmatis* mc$^2$155 cultures were grown in Middlebrook 7H9 broth or 7H10 agar complete medium (BD Difco™) supplemented with 0.2% glycerol, 10% ADC (2% glucose, 0.2% bovine serum albumin (BSA), and 0.8% NaCl), and 0.05% Tween-80 at 37 °C with shaking at 220 rpm. Overproduction of recombinant WhiB1 in *M. smegmatis* mc$^2$155 was achieved using the expression plasmid (pGS2524). The open reading frame of *whiB1* was amplified using MSWb1F and MSWb1R primers (Supplementary Table 4), and ligated into pMyNT plasmid (Addgene) between NcoI and HindIII sites, respectively. The plasmid (pGS2524) coded for WhiB1$_{MS}$ (His$_6$-TEV cleavage site-WhiB1).

The expression plasmid pGS2183 (Supplementary Table 3), which contains the open reading frame of *M. tuberculosis sigA* ligated into pET28a between NdeI and HindIII sites, was used to overproduce a His$_6$-Thrombin cleavage site-$\sigma^A$ protein in *E. coli* BL21 (λDE3). Overproduction of the C-terminal domain of $\sigma^A$ ($\sigma^A$CTD), consisting of amino acids Ala447–Asp528, was achieved using *E. coli* BL21 (λDE3) transformed with pGS2566. This plasmid contains the region corresponding to

$sigA_{G1338-A1587}$ amplified by PCR using SigAc82F and SigACTDR primers (Supplementary Table 4). The product was ligated into pET28a between NdeI and HindIII sites. The resulting plasmid coded for $\sigma^A$CTD (His$_6$-Thrombin cleavage site-$\sigma^A$CTD). The authenticity of the all plasmids was confirmed by DNA sequencing.

**Purification of WhiB1**. For isolation of the recombinant WhiB1 overproduced in *E. coli* BL21 (λDE3), cultures of the expression strain JRG6769 were grown at 37 °C in LB broth to an OD$_{600}$ ~0.7; then 1 mM IPTG was added and the cultures were incubated at 18 °C for a further 20 h before harvesting biomass by centrifugation. WhiB1 was also overproduced in *M. smegmatis* mc$^2$155 using the expression strain JRG6798. This was achieved by growing the bacterial cultures at 37 °C in Middlebrook 7H9 expression medium (not supplemented with BSA). When the OD$_{600}$ of the cultures reached ~1.0 (~24 h after inoculation), 0.2% acetamide was added to induce WhiB1 expression. Incubation was continued at 37 °C for a further 12 h before collecting biomass by centrifugation.

Overproduction of $^{15}$N- and $^{13}$C-labeled WhiB1 was achieved by growing *M. smegmatis* mc$^2$155 JRG6798 cultures at 37 °C in medium containing 0.2% glycerol-$^{13}$C$_3$ (Sigma-Aldrich) and 0.25% $^{15}$NH$_4$Cl (Sigma-Aldrich) for ~40 h to an OD$_{600}$ ~1.0 before induction with 0.2% acetamide. After further incubation at 37 °C for ~18 h, the bacteria were harvested and stored at −20 °C.

For WhiB1 purification, frozen cells (*E. coli* or *M. smegmatis*) were re-suspended in 50 mM Tris (pH 8.0) containing 0.5 M NaCl, and then lysed by sonication (3 cycles of short pulses for *E. coli*, or 10 cycles for *M. smegmatis*) using a Soniprep 150 Plus (MES) sonicator; each pulse cycle (~20 s) was followed by a pause (30 s) and the suspensions were kept on ice to avoid overheating. Lysates were cleared by centrifugation and the resulting cell-free extracts were applied to a Hi-Trap chelating column (GE Healthcare). The recombinant His$_6$-WhiB1

proteins were eluted using a linear imidazole gradient (0–500 mM). WhiB1-containing fractions were either used immediately or stored at −20 °C. For NMR experiments, buffer exchange of purified WhiB1 was carried out using a 0.5 ml Zeba™ spin desalting column (Thermo Fisher) pre-equilibrated with 25 mM $NaH_2PO_4$ (pH 7.0) containing 0.25 M NaCl.

**Co-expression of untagged WhiB1 and His₆-tagged σᴬ or σᴬCTD**. Co-expression of untagged WhiB1 and His-tagged σᴬ was achieved by transforming *E. coli* BL21 (λDE3) with pGS2560 and pGS2183 to create *E. coli* JRG6859 (Supplementary Table 3). Cultures were grown in LB broth containing ampicillin and kanamycin at 37 °C until the $OD_{600}$ reached ~0.7. To induce expression of untagged WhiB1 and the His₆-σᴬ simultaneously, IPTG was added to a final concentration of 1 mM. The cultures were incubated at 18 °C for a further ~18 h before harvesting the bacteria by centrifugation and storing the pellets at −20 °C. Purification was carried out as described above using Hi-Trap chelating chromatography. The WhiB1-σᴬCTD complex was similarly overproduced and purified using the expression strain (*E. coli* JRG6857), which carries pGS2566, coding for His₆-σᴬCTD, in place of pGS2183. For NMR experiments, overproduction and purification of ¹⁵N-labeled WhiB1-σᴬCTD complex were as described above, except cultures were grown in minimal medium prepared with ¹⁵NH₄Cl as the sole nitrogen source.

**Purification of WhiB1 and σᴬ or σᴬCTD complex**. The WhiB1-σᴬ or WhiB1-σᴬCTD complexes were enriched by Hi-Trap chelating chromatography followed by dilution with 50 mM Tris (pH 8.0) to reduce the salt concentration to 0.2 M. These fractions were then applied to a Hi-Trap Heparin HP column (GE Healthcare) and bound complexes were eluted using a linear NaCl gradient (0–1 M). Fractions containing the complexes were further purified by size exclusion chromatography using a Superdex 200 pg gel filtration column (GE Healthcare) equilibrated with 50 mM Tris (pH 8.0) containing 0.5 M NaCl. Samples of protein were analyzed after each step of purification by SDS-PAGE.

**Reaction with O₂ and NO**. Reaction of [4Fe-4S]-WhiB1 proteins with O₂ and/or NO were generally carried out in sealed Hellma 10 mm cuvettes[48]. For reaction with O₂, air-saturated buffer was injected to achieve a final concentration of ~100 μM O₂. UV-visible spectra (Cary 50 Bio UV-Vis spectrophotometer, Agilent) were obtained to monitor the reaction. For reaction with NO, a stock solution of the NO donor Proli-NONOate (Cayman Chemicals, $t_{1/2}$ = 1.8 s at 37 °C, pH 7.4) was prepared in 10 mM NaOH and the concentration was quantified optically ($\varepsilon_{252}$ 8400 $M^{-1}$ cm$^{-1}$). Holo-WhiB1 protein was titrated by progressive addition of increasing concentrations of NO by direct injection of Proli-NONOate, which had been mixed with assay buffer (50 mM $NaH_2PO_4$, 100 mM NaCl, pH 7.4) to neutralize the NaOH in the concentrated stock solution, into the protein sample via a gas-tight syringe (Hamilton). Following each NO addition, the reactions were incubated at 25 °C for 3 min before UV-visible spectra were obtained.

**Protein concentration measurement**. The concentration of reconstituted WhiB1 was determined by the method of Bradford with BSA as the standard[49]. Oxygen-saturated buffer was added to bleach the iron–sulfur cluster of WhiB1$_{EC}$ at which point apo-WhiB1 concentration was measured using the WhiB1 extinction coefficient ($\varepsilon_{280}$ 16,500 $M^{-1}$ cm$^{-1}$). A correction factor was calculated (0.79), which was then applied to subsequent measurements made by the Bradford assay.

**Iron–sulfur cluster reconstitution**. For iron–sulfur cluster reconstitution, the NifS method was used under anaerobic conditions. NifS protein was added at a ratio of 5:100 (w/w) of WhiB1, in 50 mM Tris (pH 8.0) buffer containing 0.5 M NaCl. Reconstitution was initiated by addition of 10 mM dithiothreitol (DTT), 5 mM ammonium ferrous sulfate, and 10 mM L-cysteine. The reaction was incubated at 20 °C in an anaerobic workstation for 16 h. Dialysis against 25 mM $NaH_2PO_4$ (pH 7.5), 0.25 M NaCl for ~24 h under anaerobic conditions removed unincorporated reagents. An extinction coefficient for the iron–sulfur cluster ($\varepsilon_{420}$ 16,750 $M^{-1}$ cm$^{-1}$) was used to calculate the amount of [4Fe-4S]$^{2+}$ cluster in the reconstituted protein. Iron content was determined after releasing iron by boiling a known amount of WhiB1 protein in 1% trichloroacetic acid and then adding the supernatant to a solution of the chelating agent bathophenanthroline sulfonic acid in the presence of the reducing agent ascorbic acid. The absorbance of the mixture was measured at 535 nm and the concentration of iron calculated by comparison to a standard iron solution (BDH)[20].

**Quantitative reverse transcription PCR**. Total RNA was isolated and was used with appropriate oligonucleotides designed in Primer Express (Applied Biosciences) for *espA*, *whiB1*, and *sigA* to create cDNA using a QuantiTect reverse transcription kit (Qiagen) (Supplementary Table 4). Quantitative reverse transcription PCR (qRT-PCR) was carried out on an ABI Prism 7700 instrument using the Fast SYBR Green master mix (Applied Biosystems)[50].

**Transcription analysis using DNA microarrays**. Microarray slides were scanned as previously discussed and images quantified using Bluefuse for Microarrays v3.6 (BlueGnome)[50]. Three biological replicates were performed for each condition,

carried out in duplicate for dye swaps yielding from six slides, including dye swaps, from three bacterial cultures. Data were analyzed using GeneSpring version 13 (Agilent), applying a global Lowess normalization to remove differences in dye-incorporation efficiencies between microarrays. Array features with a Bluefuse confidence of <0.1 were eliminated. Gene expression was deemed to be altered if a >threefold change in absolute expression was detected, which passed significance filtering by using a *t* test (*p* value <0.1) with a Benjamini and Hochberg multiple testing correction. The array design is available in ArrayExpress (accession number A-BUGS-23). Fully annotated microarray data have been deposited in ArrayExpress (accession no. E-MTAB 5814).

**Two-hybrid assay**. The bacterial adenylate cyclase-based two-hybrid (BACTH) system was used to detect interaction between WhiB1 and σᴬCTD in vivo[35]. The genes encoding WhiB1 and σᴬCTD were amplified by PCR. For WhiB1, pKT25-WB1F and pKT25-WB1R primers (Supplementary Table 4) were used and, after digestion with XbaI and KpnI, the product was digested and ligated into the corresponding sites of pKT25 (pGS1672) to create pGS2568. For σᴬCTD, pUT18-SigAF and pUT18-SigAR primers (Supplementary Table 4) were used and, after digestion with HindIII and KpnI, the product was ligated into the corresponding sites of pUT18 (pGS1669) to create pGS2567. The plasmids were propagated in *E. coli* K-12 (XL1-Blue) and purified (Qiagen miniprep kit). The plasmids were used to transform *E. coli* BTH101 (JRG4968). Transformants (*E. coli* JRG6862) carrying both pGS2567 and pGS2568 were selected on medium containing ampicillin (100 μg ml$^{-1}$), kanamycin (25 μg ml$^{-1}$) and streptomycin (25 μg ml$^{-1}$). Interaction of WhiB1 and σᴬCTD was measured in aerobic cultures of *E. coli* JRG6862 (Supplementary Table 3) grown in 5 ml LB broth containing ampicillin (200 μg ml$^{-1}$), kanamycin (25 μg ml$^{-1}$) and streptomycin (25 μg ml$^{-1}$) in 25 ml Sterilin tubes at 37 °C for 16 h with rocking; where indicated, cultures were supplemented with dipyridyl (0.5 mM final concentration). Anaerobic cultures were grown in LKB MES (tryptone, 10 g l$^{-1}$; yeast extract, 5 g l$^{-1}$; KCl, 6.4 g l$^{-1}$ buffered with 100 mM MES (2-(*N*-morpholino)ethanesulfonic acid) pH 5.5 supplemented with IPTG (100 μg ml$^{-1}$) and either fumarate or nitrite (1.4 mM) as indicated. Growth under the latter conditions exposes the culture to NO, generated chemically from acidified nitrite[51]. Aerobic and anaerobic control cultures, consisting of *E. coli* BTH101 transformed with pKT25 (pGS1672) and pUT18 (pGS1669) (*E. coli* JRG5386), or with pKT25-zip (pGS1673) and pUT18c-zip (pGS1671) (*E. coli* JRG5387) were also analyzed. The former strain contains only the BACTH vectors and hence acts as a negative control; the latter strain expresses fusions to the leucine zipper region of GCN4 thereby acting as a positive control[35]. β-Galactosidase activities were measured for a minimum of three independent cultures[52]. As expected, the control strain expressed low β-galactosidase activity (102 ± 13 Miller units) under the conditions tested.

**NMR methods**. ¹⁵N, ¹³C double labeled WhiB1 was concentrated to 400 μM in 25 mM sodium phosphate, 250 mM NaCl, pH 7, containing 10% D₂O. Backbone and side chain signals were assigned using a combination of triple resonance experiments, including HNCO, HN(CA)CO, HNCA, HN(CO)CA, HNCACB, CBCA(CO)NH, HNCA(N)NH, HCCH-TOCSY, CCH-TOCSY, HBCB(CGCD)HD, HBCB(CDCGCE)HE, ¹³C HSQC centered on aromatic signals, and homonuclear NOESY and TOCSY experiments. Some additional assignments of very rapidly relaxing signals were made using fast HSQC, with a recycle delay of 0.3 s, acquisition time of 17 ms, and 1/4 J INEPT transfer delays of 833 μs. NOESY spectra used simultaneous acquisition of ¹³C and ¹⁵N with a mixing time of 100 ms. The $R_1$ relaxation rates of amide protons were measured using standard experiments, plus an experiment optimized for rapid relaxation[53]. Relaxation delays of 1, 10, 30, 60, 110, 200, 400, 800, and 1200 ms were used, and relaxation times were obtained by fitting to an exponential decay using home-written scripts. Structure calculations were carried out using crystallography and NMR system (CNS) with standard torsion angle refinement[54]. The backbone chemical shift assignments were used to generate φ, ψ, and χ₁ dihedral angle restraints using TALOS-N[55]. Further restraints were obtained from NOESY spectra (using restraints grouped into strong (<3 Å), medium (<4 Å), and weak (<5 Å)). In addition, nuclei that could not be observed in NMR spectra (N of Val8, Cys9, Cys37, Val42, Thr43, Gly61, and Gly62; CA of Cys9, Cys46, and CB of Cys9, Glu45, Cys46) were restrained to be within 6.0 Å of the center of the cluster; and nuclei with $T_1$ values of <50 ms (N of Ala7, Arg10, Asn38, Cys40, Cys46, Met63) were restrained to be within 6.5 Å of the center of the cluster. A small number of nuclei (CB of Arg10, Glu12, Lys6; CG of Val8, Arg10, Glu12, Lys6, Thr43; CD of Arg10 and Lys6) that were not noticeably broadened were restrained to be more than 6.5 Å from the center of the cluster to improve convergence of the calculation. Hydrogen bonds were added when both TALOS-N and Phyre predicted the location of α-helices[26]. The cluster was represented by a single (nominally) zinc atom at the center of the cluster, restrained to have typical [4Fe-4S] geometry, i.e., with Zn-Cys S distances of 3.9 Å and S-Zn-S angles of 110°, which were encoded into the zn2.top, parallhdg5.3Z.pro, and topallhdg5.3z.pro files. Structures were visualized in PyMOL (The PyMOL Molecular Graphics System, Version 1.8 Schrödinger, LLC). The WhiB1-His-σᴬCTD complex was expressed and purified as a complex, meaning that chemical shift assignments for WhiB1 could not be made by following titration shifts. The shift changes were therefore determined by measuring the smallest shift differences between an assigned WhiB1 signal and an unassigned signal in the complex, the so-called "minimum chemical shift procedure[56]".

The final WhiB1 structural parameters are provided in Supplementary Table 5. A Ramachandran analysis of the 10 lowest energy structures (Supplementary Fig. 16) was carried out using MolProbity[57] and showed 92.2% of residues in the most favored regions, and 3.9% in the additionally allowed regions.

**Electrophoretic mobility shift assay**. To investigate the ability of WhiB1 to bind at the promoter region of *espACD* (P*espA*), this region was divided into five fragments (F1, F2, … F5), where F1 is the nearest to *espA*, while F5 is the nearest fragment to *ephA* (Supplementary Fig. 13). The five fragments were amplified separately by PCR using (pGS2314), which contains the genomic non-coding area of *espACD* as a template. Each fragment was generated by using a forward non-biotin-labeled DNA primer, and a reverse-labeled primer (eurofins genomics). Each fragment was 270 bp (except F5; 277 bp). PCR products were purified from agarose gels using QIAquick Gel Extraction Kit (Qiagen). Apo-WhiB1 protein was reduced with 1 mM DTT and incubated on ice for at least 1 h before mixing the protein with DNA. Various concentrations of the protein (5, 10, and 15 μM) were incubated with 15 fmol of 5′-biotin-labeled DNA in EMSA buffer (25 mM NaH$_2$PO$_4$ (pH 7.5), 200 mM NaCl, and 5% glycerol). Samples were incubated for 15 min at room temperature. A 200-fold excess of unlabeled non-specific DNA or 150-fold excess of unlabeled specific DNA competitors were also used. For competition, the protein was incubated first with unlabeled (specific or non-specific) DNA competitor for 15 min before adding the labeled DNA. Loading buffer 6× (15% Ficoll 400, 0.25% Bromophenol blue, 0.25% Xylene cyanol, and 1× TBE) was added to the samples prior to loading onto 7.5% polyacrylamide Tris-glycine gels. After electrophoresis, DNA was transferred to a nylon membrane (GE Healthcare), and crosslinked using UV-light (120 mJ cm$^{-2}$ for ~90 s). The chemiluminescent nucleic acid detection module (Thermo Scientific) was used to detect biotin-labeled DNA according to the manufacturer's instructions.

**Mass spectrometry**. For ESI-MS under non-denaturing conditions, an aliquot of [4Fe-4S] WhiB or WhiB1-σ$^A$CTD complex was exchanged into 250 mM ammonium acetate pH 8.0 using a midi-PD10 desalting column (GE Healthcare). The volume of the eluent was increased to 1.6 ml and the concentration of [4Fe-4S] WhiB1 determined via absorbance at 406 nm. Initial experiments with aqueous solutions of NO gas resulted in the suppression of protein ionization. Therefore, the slow release agent diethylamine (DEA) NONOate was used to study the effects of NO on the WhiB1-σ$^A$CTD complex. DEA NONOate (Sigma-Aldrich) solutions were prepared immediately before use in cold (~4 °C) ammonium acetate buffer, and quantified by absorbance ($ε_{250nm} = 6500$ M$^{-1}$ cm$^{-1}$). At 25 °C, DEA NONOate (2.7 mM), which decayed with a half-life of 17 min in the ammonium acetate buffer to yield 1.5 mol NO per NONOate, was pre-decayed for ≥2.5 h to give a saturated NO solution (~1.75 mM), prior to mixing with the WhiB1:σ$^A$CTD complex. Briefly, an aliquot (up to 20 μl; see figure legends for further details) of decayed DEA NONOate was added directly to a sample (200 μl) of the WhiB1:σ$^A$CTD complex to give a specific ratio of NO to [4Fe-4S] cluster. Samples were immediately loaded into a 500 μl gas-tight syringe (Hamilton), and infused directly using a syringe pump (0.3 ml h$^{-1}$) into the ESI source of a Bruker micrOTOF-QIII mass spectrometer (Bruker Daltonics, Coventry, UK) operating in the positive ion mode. For NONOate-treated samples, the syringe was thermostatically maintained at 25 °C. Control experiments in which the WhiB1:σ$^A$CTD complex was mixed with diethylamine, the breakdown product of NONOate decomposition, confirmed that observed reaction was entirely due to reaction with the released NO. Similar MS experiments were performed with *S. coelicolor* [4Fe-4S].

MS data for [4Fe-4S] WhiB1 and [4Fe-4S] WhiD were acquired using Bruker oTOF Control software over the *m/z* range 500–3000, with parameters as follows: dry gas flow 4 l min$^{-1}$, nebulizer gas pressure 0.4 Bar, dry gas 180 °C, capillary voltage 4500 V, offset 500 V, ion energy 5 eV, collision RF 200 Vpp, and collision cell energy 10 eV. For the WhiB1:σ$^A$CTD complex and DEA NONOate-treated samples, data were acquired over the *m/z* range 1300–3500, with the following parameters: dry gas flow 4 l min$^{-1}$, nebulizer gas pressure 0.8 Bar, dry gas 180 °C, capillary voltage 3750 V, offset 500 V, ion energy 4 eV, quadrupole isolation mass of 1300 or 2000 *m/z*, collision RF 650 Vpp, collision cell energy 6 eV[58]. To investigate cluster and complex disassembly, incremental increases to the in-source collision-induced dissociation (isCID; equivalent to the cone voltage) were applied over the range of 0–140 eV. The ESI-TOF mass spectrometer was calibrated with the ESI-L low concentration tuning mix provided by Agilent Technologies (San Diego, CA). The gas-tight injection syringe (Hamilton) and the associated tubing (PEEK tubing, Upchurch Scientific, which has low O$_2$ permeability − ~14 ml O$_2$ per 250 cm$^2$ at 25 °C and 1 atmosphere in 24 h) were flushed with anaerobic ammonium acetate buffer (5 ml) before sample application. Data were processed by Compass DataAnalysis software version 4.1 (Bruker Daltonik, Bremen, Germany) with neutral mass spectra (between 9000 and 22,000 Da) created using ESI Compass version 1.3 Maximum Entropy deconvolution software generating isotope average neutral masses. For apo-proteins, the peaks correspond to the $[M + nH]^{n+}/n$ species. For proteins that retain an iron–sulfur cluster or degradation products thereof, additional charge is contributed by the cluster and therefore the peaks correspond to $[M + FeS^{x+} + (n-x)H]^{n+}/n$, (*M*, protein mass; FeS, mass of iron–sulfur cluster with charge *x*+; *H*, mass of a proton; and *n*, the total charge)[30]. Predicted masses are isotope averages for neutral proteins or protein complexes, in which cofactor binding is charge compensated[29].

The *S. coelicolor* WhiD protein was overproduced as a (His)$_6$-tag fusion in aerobic *E. coli* cultures (BL21 λDE3 star transformed with expression plasmid pIJ6631) and applied to a nickel-charged HisTrap chelating column (GE Healthcare) equilibrated with 50 mM Tris-HCl, 5% glycerol, 250 mM NaCl, pH 7.3. After washing with 50 mM Tris-HCl, 100 mM NaCl, 50 mM imidazole, 5% glycerol, pH 7.3 to remove non-specifically bound proteins, the His-tagged WhiD was recovered by applying a linear gradient of 50–500 mM imidazole in 50 mM Tris-HCl, 100 mM NaCl, 5% glycerol, pH 7.3[8].

**Far UV circular dichroism spectroscopy**. A Jasco J810 spectropolarimeter was used to analyze WhiB1$_{MT}$ (30 μM) in 20 mM sodium phosphate buffer, pH 7.4 containing 0.1 M NaCl at 25 °C.

**Data availability**. The WhiB1 NMR data and structure coordinates are available in BMRB: 34153 and PDB: 5OAY. The gene expression data are available in ArrayExpress: E-MTAB-5814. Other data are available from the corresponding author on reasonable request.

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

## Acknowledgements

This work was supported by Biotechnology and Biological Sciences Research Council UK project grants BB/K000071/1 and BB/L008114/1 (J.G.), BB/L007673/1 and BB/P006140/1 (J.C.C./N.E.L.B.), Medical Research Council grant U117585867 (R.S.B.), and a Higher Committee for Education and Development in Iraq PhD scholarship (B.K.K.). We thank Dr S. Sedelnikova (University of Sheffield) for protein purification advice, University of East Anglia for funding the Q-TOF instrument and the late Prof. P.J. Artymiuk for his contributions during the early stages of this work.

## Author contributions

J.G., L.J.S., M.P.W., N.E.L.B., and R.S.B. conceived the study and supervised the experiments. A.M.H., B.K.K., D.M.H., J.C.C., and M.D.R. carried out the experiments; all authors contributed to analyzing data and writing the manuscript.

## Additional information

**Competing interests:** The authors declare no competing financial interests.

