## [Peer Review File · Nature Communications]

Reviewers' comments:

Reviewer #1 (Remarks to the Author):

The manuscript by Kudhair et al. reports on the structural characterization of the WhiB1 protein from *Mycobacterium smegmatis* and on the analysis of its interactions with the sigma factor and the role of NO on its properties.

The characterization is detailed and the experiments carefully performed. The topic is relevant but the manuscript needs extensive revision.

The main points are:

1- The nature of the cluster should be better characterized. EPR spectra could contribute to confirm its nature.

2- The structure is determined without a detailed characterization and description of the cluster binding region. Structure calculations with the inclusion of the cluster and of some structural info on the surrounding of the cluster should be performed.

3- Pg 4 lines 124-125 "...the structure suggests that...": to proof this hypothesis the authors should analyze the structural features of the apo state, at least by just comparing the spectra of the two forms, showing that indeed the residues of the C-terminal helix show marked differences.

4- The authors should describe the features of the various sulfur adducts of the protein.

Minor points:

Pg 3 line 105 Add some references to the most recent literature on NMR experiments for detecting fast relaxing signals, in particular in Fe-S proteins.

Pg 4 line 112 "Accordingly" seems not appropriate. There is no relation with the previous sentence.

Pg 4 line 119 "dynamically averaged": what does it mean?

Finally, the text should be made more concise so that the overall message of the work becomes easier to get.

Reviewer #2 (Remarks to the Author):

In this manuscript, the authors presented the first NMR structural model of WhiB1 from *Mycobacterium tuberculosis*. They also reported that WhiB1 purified from *M. smegmatis* contains a stable [4Fe-4S] cluster. While WhiB1 forms a protein complex with the major sigma factor A, loss of the [4Fe-4S] cluster in WhiB1 by nitric oxide treatment releases the sigma factor A from WhiB1. The authors proposed that nitric oxide converts the WhiB1 [4Fe-4S] cluster to apo-WhiB1, which will release the major sigma factor A from WhiB1 and trigger a major reprogramming of gene expression in *M. tuberculosis*. The results could be interesting and highly significant.

There are two concerns that the authors might consider:

First is the mechanism by which the WhiB1 [4Fe-4S] cluster is converted to apo-WhiB1 by nitric oxide. In Figure 5f, less than 10% of the WhiB1 [4Fe-4S] cluster was apparently changed to apo-WhiB1 after nitric oxide treatment. While some of the WhiB1 [4Fe-4S] cluster were converted to apo(S)-WhiB1 or apo(S)₂-WhiB1 (Figure 5), most of the WhiB1 [4Fe-4S] cluster appeared to form a big protein complex after nitric oxide treatment (Figure 2C). What is the identity of the "big protein complex"? Could the "big protein complex" be eventually released as apo-WhiB1 and sigma factor A? If so, what could be the underlying mechanism?

Second is the DNA binding activity of apo-WhiB1. The authors stated that the C-terminal helix of WhiB1 has the DNA binding activity when the [4Fe-4S] cluster is lost (page 6, line 213). However, there was no evidence showing the DNA binding activity of apo-WhiB1. Would it be possible to

demonstrate the specific role of the [4Fe-4S] cluster in the WhiB1's DNA binding activity?

Reviewer #3 (Remarks to the Author):

Summary:

In this manuscript, the authors further characterize WhiB1, a wbl-transcription factor in *Mycobacterium*. Importantly, the authors generate the first structural model of a wbl protein. Using NMR, they reveal for the first time that wbl proteins are four helix bundle proteins, which are held together by the CXXC motif, which is important for the proteins to sense the environment. This work is impressive in its scope, and builds directly on previous work by these groups. Using ESI MS the authors confirm that conclusion that the Fe-S cluster in WhiB1 is NO-sensitive and O₂-stable. It was previously shown that WhiB1 interacts with σ^a . The authors showed that they could co-express WhiB1 and σ^a in *E. coli*; pulling down σ^a resulted in co-IP of WhiB1, but only when WhiB1 could acquire an iron-sulfur cluster. Thus, the iron sulfur cluster is essential for interaction with σ^a . The authors then go on to show that interaction of the WhiB1 Fe-S cluster with NO results in a dissociation of σ^a from WhiB1, using size exclusion chromatography, SDS page and absorbance spectroscopy. It was previously shown that other Wbl proteins bind region 4 of σ^a . Using Co-IP from their co-expression system in *E. coli*, WhiB1 also interacts with region 4 of σ^a . Interestingly, they showed that the DNA binding domain does not interact with σ^a , rather, the region that is adjacent to the Iron sulfur cluster interacts with σ^a CTD. Using a two-hybrid approach, the authors showed that restricting iron and exposure to NO of *E. coli* lead to dissociation of WhiB1 and σ^a as measured by the two hybrid system. Together, this part of the manuscript contributes a deeper understanding of a fundamental transcriptional mechanism that likely important outside of mycobacteria as well. In addition to contributions to the mechanism of transcriptional regulation, and how exposure to the environment impacts the interaction of WhiB1 and σ^a , the authors provide first insight into the genes regulated by WhiB1 in mycobacterium. The authors, finally, generate a series of strains in which WhiB1 expression is reduced (since the gene is essential). They do this in a clever manner, by deleting parts of the whiB1 promoter. Using the lowest whiB1 expressing strain, the authors identify transcripts regulated by whiB1. They demonstrate that WhiB1 is an activator and a repression of gene expression. The operon with the biggest change in expression was the *espACD* operon, which encodes the EspA and EspC ESX-1 substrates. Using qRT-PCR the authors show that reduction of *whiB1* expression leads to an increase in *espA* transcript. They then go on to present a model to put their work in the context of mycobacterial virulence.

Major Comments:

I think overall, the authors provide strong evidence to back most of their conclusions. The use of the heterologous models was clever. I think the biggest weakness is when the authors try to put their findings in the context of virulence. I think the model at the end of the paper would be much stronger if the authors had tested the virulence and secretion phenotypes for the strains expressing intermediate levels of *whiB1* expression and therefore increased levels of *espACD* expression. I do not ask for this lightly, as I understand the work required. However, it is unclear how **increasing** expression of *espACD* 4-fold impacts ESX-1 secretion and virulence in *M. tuberculosis*. Understanding the meaning of this finding in virulence would make the story interesting to a wider audience, capturing the interest of investigators in the ESX fields and interested in mycobacterial pathogenesis. At the very least, it would be interesting to see how increased *espACD* impacts virulence in a macrophage model of infection, and how it impacts cytokine induction.

An alternative to this would be to focus this manuscript on the mechanism of WhiB1, through the interaction with σ^a and how environmental stress impacts this interaction, and remove the transcriptomic experiments, for the next manuscript, where a deeper focus could be made on how WhiB1 impacts virulence.

The study would be of interest to investigators studying wbl proteins in diverse disciplines. I think the authors do a very nice job of moving the field forward. They contribute fundamental insight into how wbl proteins sense the environment, interact with σ^a and importantly, how the two are connected. I think these findings would definitely influence how investigators think about wbl proteins and how they regulate transcription. I am not sure how it will influence the group of investigators studying mycobacterial virulence.

Minor Comments:

A previous report by Argarwal et al (PMID: 16946269) and by the authors of this study (PMID:20929442) indicated that the whiB1 promoter in *M. tb* was regulated by CRP (Rv3676). Considering that the *espA* promoter is also regulated directly by CRP, the authors should discuss how the regulation of *espA* by WhiB1 could be influenced by CRP. This lends another level of regulation to an already tightly regulated operon.

The notation of WhiB1MS for WhiB1 from *M. tuberculosis* being produced in *M. smegmatis* is confusing. In ESX nomenclature, it is commonly used as follows: if the protein is from *M. smegmatis*, you use the MS subscript. If it is from TB, you use the MT subscript. Considering that your audience will include ESX researchers, this should be changed

Reviewer #1

This reviewer comments that 'The characterization is detailed and the experiments carefully performed.' and raises four points.

1- The nature of the cluster should be better characterized. EPR spectra could contribute to confirm its nature.

Despite early reports that refolded Wbl proteins may bind a [2Fe-2S] cluster, it is now well established that Wbl proteins, including *M. tuberculosis* WhiB1 and WhiB3 and *S. coelicolor* WhiD, contain a [4Fe-4S] cluster. A wide range of biophysical methods, including optical and EPR spectroscopies have been applied (Jakimowicz et al. 2005 J. Biol. Chem., 280, 8309-15; Singh et al. 2007 Proc. Natl. Acad. Sci. USA 104, 11562-7; Crack et al. 2009 Biochemistry 48, 12252-64; Smith et al. 2010 Biochem. J. 432, 417-27; Crack et al. 2011 J. Am. Chem. Soc. 133, 1112-1121; Serrano et al. 2016 Angew Chem. Int. Ed. 55, 14575-14579). For example, UV-visible and CD data for WhiB1 are entirely consistent with it being a [4Fe-4S], and the EPR spectrum of the [4Fe-4S] form was featureless, consistent with the cluster being in the EPR-silent diamagnetic [4Fe-4S]²⁺ state. Attempts to reduce the cluster to an EPR active form with the strong reductant dithionite resulted in bleaching of the sample, indicating that the cluster was lost (Smith et al. 2010 Biochem. J. 432, 417-27).

The UV-visible, CD and EPR properties of the iron-sulfur cluster bound form of WhiB1 are extremely similar to those of *S. coelicolor* WhiD, for which further spectroscopic evidence is available. The resonance Raman spectrum of WhiD in the iron-sulfur stretching region (250 to 450 cm⁻¹) is highly characteristic of a [4Fe-4S]²⁺ cluster with complete cysteinyl ligation (Crack et al. 2009 Biochemistry 48, 12252-12264). More recently, the application of the novel iron-specific technique nuclear resonance vibrational spectroscopy (NRVS) to WhiD also demonstrated the presence of a [4Fe-4S] cluster (Serrano et al. 2016 Angew Chem. Int. Ed. 55, 14575-14579).

Here, we present further evidence, from ESI-MS, that WhiB1 contains a [4Fe-4S] cluster. We have recently analysed a number of iron-sulfur cluster regulatory proteins using ESI-MS, and have found that it is extremely good for determining the type of cluster present (for example, see recent papers on [4Fe-4S] FNR, [4Fe-4S] NsrR, and [2Fe-2S] RsrR (Crack et al. 2017, Proc. Natl. Acad. Sci. USA 114, E3215-E23; Volbeda et al. 2017, Nat. Commun. 8, 15052; Munnoch et al. 2016, Sci. Reports 6, 31597). The cluster types bound by NsrR and RsrR has been independently confirmed via the crystal structure (with the Fontecilla-Camps group, Volbeda et al. 2017, Nat. Commun. 8, 15052 and unpublished data). Our work follows that on ferredoxins (Johnson et al. 2000, Anal. Chem. 72, 1410-8), which first established the ability of mass spectrometry to determine cluster type.

Thus, we believe that it is established beyond doubt that Wbl proteins, including *M. tuberculosis* WhiB1, contain a [4Fe-4S]²⁺ cluster. We have added text to summarize the evidence that Wbl proteins possess well characterized [4Fe-4S] clusters (lines 141-142).

2- The structure is determined without a detailed characterization and description of the cluster binding region. Structure calculations with the inclusion of the cluster and of some structural info on the surrounding of the cluster should be performed.

We now include an additional panel in Figure 1 showing the environment of the iron-sulfur cluster accompanied by new text to describe the features thereof (lines 109-117). The structural calculations did include the cluster and the surroundings of the cluster. Specifically, the calculation included sulfur-sulfur distances from the cluster ligands, and a fixed geometry for the cluster itself, including angular information on ligation of the cluster. It also used experimental information on distances from the cluster to surrounding residues as structural restraints. Details are provided in the Methods section.

3- Pg 4 lines 124-125 "...the structure suggests that...": to proof this hypothesis the authors should

analyze the structural features of the apo state, at least by just comparing the spectra of the two forms, showing that indeed the residues of the C-terminal helix show marked differences.

We thank the referee for this suggestion. We now include some data on the structure of the apo-protein, shown in Extended Data Fig. 5 and discussed in the text. CD spectra show that on removal of the cluster, most of the helical structure is lost, with only a small proportion of helix remaining. The NMR spectrum of the apo-protein shows some interesting features. It is very different from the holo-protein, with most signals moving towards a more random coil position (i.e. closer to the ^1H range 7.5–8.5 ppm, which is where random coil signals typically lie). This shows that most of the protein is now close to random coil. However, about 15 signals are missing. This suggests dynamic processes occurring on an intermediate timescale (i.e. roughly ms to μs), involving a significant fraction of the protein, thereby implying that some parts of the protein retain structure. Only about 12 signals in the HSQC spectrum of the apo-protein remain at positions close to where they are in the holo-protein. These include a number of residues in the disordered termini, but significantly also include residues 69, 73, 75, and 78. These four residues are the most outward facing residues within the C-terminal helix, and strongly imply that the only part of the apo-protein that retains anything like its native conformation is the C-terminal helix. This is exactly what one would hope to find if our hypothesis is true, that in the apo-protein the C-terminal helix is free to interact with DNA, and this observation therefore offers powerful support to the hypothesis. It could also be speculated that interaction of apo-WhiB1 with DNA contributes to stabilizing the apo-conformation; this is not without precedent (e.g. SASPs become α -helical only when bound to DNA; Hayes et al. 2000 J. Biol. Chem. 275, 35040-35050). We have included extra text covering these observations (lines 122-135).

4- The authors should describe the features of the various sulfur adducts of the protein.

The ESI-MS data shows that sulfur adduct species containing one and two sulfurs per apo-protein are formed via in source collision induced dissociation (isCID) and following nitrosylation of [4Fe-4S] WhiB1. Previous studies of WhiD showed that cluster sulfide is oxidised to sulfane (S^0) during nitrosylation (Crack et al 2011 J. Am. Chem. Soc. 133, 1112-21) and sulfur adducts were also reported following the nitrosylation of [4Fe-4S] FNR, as well as following reaction with O_2 (Crack et al. 2013 J. Biol. Chem. 288, 11492-502; Crack et al 2017, Proc. Natl. Acad. Sci. USA 114, E3215-E23). In the latter case, this stored sulfane sulfur was shown to be available to rebuild an FeS cluster, *in vitro*, without the requirement for a cysteine desulfurase (Zhang et al. 2012 Proc. Natl. Acad. Sci. USA 109, 15734-9); such a process could have biological significance. Thus, it appears that sulfide oxidation is a common process (and not a novel observation here). The WhiB1 sulfide adducts are present as Cys persulfides, but we do not know at this stage which Cys residues are modified. We have made minor modifications to the relevant text to make this clear (lines 163-165).

Minor points:

Pg 3 line 105 Add some references to the most recent literature on NMR experiments for detecting fast relaxing signals, in particular in Fe-S proteins.

We have added references to the experiments that we used, which date from 1994 together with a more recent paper (2006) describing recent strategies (Refs 24 and 25).

Pg 4 line 112 "Accordingly" seems not appropriate. There is no relation with the previous sentence.

We have modified the text.

Pg 4 line 119 "dynamically averaged": what does it mean?

We adopted this terminology from the TALOS-N output, but have modified it to make more sense. The chemical shifts are close to random coil, meaning that this region is almost completely disordered.

Finally, the text should be made more concise so that the overall message of the work becomes easier to get.

We have made some changes to the text to attempt to address this point, mainly in shortening the first results section by cutting the description of the analysis of WhiB1_{MT} expressed in *E. coli* (lines 81-86). We believe that the overall message of our work is clearly expressed in the first paragraph and in the revised Fig. 6.

Reviewer #2

This reviewer comments 'The results could be interesting and highly significant.' and raises 2 points for consideration.

First is the mechanism by which the WhiB1 [4Fe-4S] cluster is converted to apo-WhiB1 by nitric oxide. In Figure 5f, less than 10% of the WhiB1 [4Fe-4S] cluster was apparently changed to apo-WhiB1 after nitric oxide treatment. While some of the WhiB1 [4Fe-4S] cluster were converted to apo(S)-WhiB1 or apo(S)2-WhiB1 (Figure 5), most of the WhiB1 [4Fe-4S] cluster appeared to form a big protein complex after nitric oxide treatment (Figure 2C). What is the identity of the "big protein complex"? Could the "big protein complex" be eventually released as apo-WhiB1 and sigma factor A? If so, what could be the underlying mechanism?

The reviewer makes a good point about the ESI-MS and the gel filtration data and we can see that we need to address this further. Firstly, the ESI-MS is not a quantitative technique, so the relative intensities cannot be directly interpreted in terms of percentage of the sample. This is because some species ionise better than others. So, while the peaks due to apo-WhiB1 forms amount to only ~20% relative abundance (relative to the starting abundance of the WhiB1-SigA complex), this does not mean that they represent ~20% of the WhiB1 protein. An illustration of this is that the apo-WhiD fraction (~30% via biochemical assays) of *S. coelicolor* WhiD samples (~70% replete with cluster) does not readily ionize in the presence of [4Fe-4S] WhiD, see Extended Data Fig. S6 black line, and is thus observed to have a relative abundance of ≤5% compared to [4Fe-4S] WhiD. A similar phenomenon may also occur with WhiB1. So, while absolute concentrations cannot be determined, changes in the intensity of the peaks can be interpreted quantitatively (e.g. Crack et al. 2017 Proc. Natl. Acad. Sci. USA 114, E3215-E23); hence, we were able to plot changes as a function of NO per cluster. We now note that ESI-MS is not quantitative for the reasons stated above.

Thus, the data do not imply that there is an unaccounted for majority of apo-WhiB1 'elsewhere'. Indeed, higher mass species were not detected in the MS experiment. However, the reviewer is right that the gel filtration data implies that reaction of NO generates a 'big protein complex', in that there is a large protein component eluting at apparently higher mass following nitrosylation of the WhiB1:SigA complex. We do not fully understand this. Iron-nitrosyl species of WhiB1 were not detected by ESI-MS and this could be due to the competing effects of the high concentration acetate buffer that was used or an ionization problem. The SDS-PAGE analysis of the fractions eluting from the size exclusion column show that the apparently high molecular weight is composed of both WhiB1 and SigA and the UV-visible spectrum is consistent with the WhiB1 iron-sulfur cluster being nitrosylated (revised Fig. 2). Therefore, we offer what is probably the simplest explanation that it represents a nitrosylated WhiB1:SigA complex that breaks down to yield apo-WhiB1 and SigA. It could be speculated that this complex might elute anomalously because SigA is an elongated molecule and nitrosylated WhiB1 induce conformation changes that enhance this as a means to promote disassociation from core polymerase, but the underlying

mechanism is unknown at present. Revised Fig. 2 and the accompanying text address these points.

Second is the DNA binding activity of apo-WhiB1. The authors stated that the C-terminal helix of WhiB1 has the DNA binding activity when the [4Fe-4S] cluster is lost (page 6, line 213). However, there was no evidence showing the DNA binding activity of apo-WhiB1. Would it be possible to demonstrate the specific role of the [4Fe-4S] cluster in the WhiB1's DNA binding activity?

We have previously compared DNA binding (EMSA and footprinting data) by apo- and holo-WhiB1, including the effect of NO and shown that the apo- and NO-treated WhiB1, but not holo-WhiB1 bind DNA (Smith et al. 2010 Biochem. J. 432, 417–27). We have also shown that positively-charged residues in the C-terminal region are required for DNA-binding by apo-WhiB1 (Smith et al. 2012 PLoS One e40407). These citations are now made in the revised text. In this manuscript we show binding of apo-WhiB1 to the *espA* promoter in Extended Data Fig. 13.

Reviewer #3

This reviewer comments 'The study would be of interest to investigators studying wbl proteins in diverse disciplines. I think the authors do a very nice job of moving the field forward. They contribute fundamental insight into how wbl proteins sense the environment, interact with σ^a and importantly, how the two are connected. I think these findings would definitely influence how investigators think about wbl proteins and how they regulate transcription.'

Major Comments:

I think overall, the authors provide strong evidence to back most of their conclusions. The use of the heterologous models was clever. I think the biggest weakness is when the authors try to put their findings in the context of virulence. I think the model at the end of the paper would be much stronger if the authors had tested the virulence and secretion phenotypes for the strains expressing intermediate levels of *whiB1* expression and therefore increased levels of *espACD* expression. I do not ask for this lightly, as I understand the work required. However, it is unclear how increasing expression of *espACD* 4-fold impacts ESX-1 secretion and virulence in *M. tuberculosis*. Understanding the meaning of this finding in virulence would make the story interesting to a wider audience, capturing the interest of investigators in the ESX fields and interested in mycobacterial pathogenesis. At the very least, it would be interesting to see how increased *espACD* impacts virulence in a macrophage model of infection, and how it impacts cytokine induction.

An alternative to this would be to focus this manuscript on the mechanism of WhiB1, through the interaction with σ^a and how environmental stress impacts this interaction, and remove the transcriptomic experiments, for the next manuscript, where a deeper focus could be made on how WhiB1 impacts virulence.

We now note at the appropriate point in the text that Fortune et al. (2005 Proc. Natl. Acad. Sci. USA 102, 10676-10681) demonstrated the correlation between *espA* expression and ESX-1 secretory activity. Hence repression of the *espA* operon by WhiB1 would result in lower ESX-1 activity in response to host NO. As the effector proteins secreted by ESX-1 are highly immunogenic it can be argued that it would be advantageous for the bacterium to switch off this system after phagocytosis and NO could be one of the signals to trigger such a response. It is known that ESX-1 is expressed early in mouse infection models and then switched off (Lazarevic et al. 2005 J. Immunol. 175, 1107-1117). We believe that the transcriptomic data are an important component of our manuscript because they demonstrate that the influence of WhiB1 on *M. tuberculosis* gene expression is not merely a minor player regulating a few genes but its influence is exerted over a broad regulon. We have revised the text and Fig. 6 to include some of these points.

Minor Comments:

A previous report by Argarwal et al (PMID: 16946269) and by the authors of this study

(PMID:20929442) indicated that the whiB1 promoter in *M. tb* was regulated by CRP (Rv3676). Considering that the *espA* promoter is also regulated directly by CRP, the authors should discuss how the regulation of *espA* by WhiB1 could be influenced by CRP. This lends another level of regulation to an already tightly regulated operon.

We have modified Figure 6 and the associated text to incorporate these points.

The notation of WhiB1MS for WhiB1 from *M. tuberculosis* being produced in *M. smegmatis* is confusing. In ESX nomenclature, it is commonly used as follows: if the protein is from *M. smegmatis*, you use the MS subscript. If it is from TB, you use the MT subscript. Considering that your audience will include ESX researchers, this should be changed.

We have revised the manuscript to avoid the need for this nomenclature.

REVIEWERS' COMMENTS:

Reviewer #1 (Remarks to the Author):

The authors have properly addressed the various points raised and significantly improved the manuscript which can be now considered for publications.